# Empirical methods for the estimation of Southern Ocean $CO_2$: Support Vector and Random Forest Regression

Luke Gregor[1,2], Schalk Kok[3], and Pedro M. S. Monteiro[1]

[1]Southern Ocean Carbon-Climate Observatory (SOCCO), CSIR, Cape Town, South Africa
[2]University of Cape Town, Department of Oceanography, Cape Town, South Africa
[3]University of Pretoria, Department of Mechanical and Aeronautical Engineering, Pretoria, South Africa

*Correspondence to:* Luke Gregor (luke.gregor@uct.ac.za)

**Abstract.** The Southern Ocean accounts for 40% of oceanic $CO_2$ uptake, but the estimates are bound by large uncertainties due to a paucity in observations. Gap filling empirical methods have been used to good effect to approximate $pCO_2$ from satellite observable variables in other parts of the ocean, but many of these methods are not in agreement in the Southern Ocean. In this study we propose two additional methods that perform well in the Southern Ocean: Support Vector Regression (SVR) and Random Forest Regression (RFR). The methods are used to estimate $\Delta pCO_2$ in the Southern Ocean based on SOCAT v3, achieving similar trends to the SOM-FFN method by Landschützer et al. (2014). Results show that the SOM-FFN approach outperforms the RFR and SVR methods with respective RMSE scores of 14.84, 16.45 and 24.40 µatm. However, this is, in part, due to an increase in coastal observations from SOCAT v2 to v3. The success of the SOM-FFN and RFR both depend on the ability to adapt to different modes of variability. The SOM-FFN achieves this by having independent regression models for each cluster, while this flexibility is intrinsic to the RFR method. Analyses of the estimates shows that the SVR and RFR's respective sensitivity and robustness to outliers define the outcome significantly. Further analyses on the methods were performed by using a synthetic dataset to assess: which method (RFR or SVR) has the best performance?; what the effect of using time, latitude and longitude as proxy variables is on $\Delta pCO_2$?; and what is the impact of the sampling bias in the SOCAT v3 dataset on the estimates? We find that while RFR is indeed better than SVR, the ensemble of the two methods outperforms either one, due to complementary strengths and weaknesses of the methods. Results also show that for the RFR and SVR implementations, it is better two include coordinates as proxy variables as RMSE scores are lowered and the phasing of the seasonal cycle is more accurate. Lastly we show that there is only a weak bias due to undersampling. The synthetic data provides a useful framework to test methods in regions of sparse data coverage and showing potential as a useful tool to evaluate methods in future studies.

## 1 Introduction

The global oceans have played an important role in mitigating the effects of climate change by taking up 25% of anthropogenic $CO_2$ emissions annually (Khatiwala et al., 2013; Le Quéré et al., 2016). The Southern Ocean has played a disproportionate role in this uptake, accounting for 40% of the oceanic anthropogenic $CO_2$ uptake (Khatiwala et al., 2013; Frölicher et al., 2015). Yet, despite the region's importance, first order $CO_2$ flux estimates are bound by large uncertainties due to sparse observations

in the Southern Ocean (Lenton et al., 2006; Monteiro, 2010; Lenton et al., 2012; Takahashi et al., 2012; Bakker et al., 2016). These uncertainties limit our capacity to resolve variability and trends of $CO_2$.

Viable alternative methods to estimate net $CO_2$ flux are atmospheric $CO_2$ inversions, ocean biogeochemical process models and empirical models (Rödenbeck et al., 2015). As shown by Le Quéré et al. (2007), atmospheric $CO_2$ inversions are useful tools to estimate the net $CO_2$ fluxes, but fail to offer further understanding with spatially integrated air-sea flux estimates (Fay and McKinley, 2014). Conversely, ocean biogeochemical process models are good tools for mechanistic understanding, but fail to represent seasonality of $CO_2$ fluxes in the Southern Ocean (Lenton et al., 2013; Mongwe et al., 2016). Empirical modelling offers an opportunity to bridge the gap between sparse data in the Southern Ocean and correct parameterisation of future earth systems models.

Empirical models maximise the utility of existing surface ocean $CO_2$ observations ($pCO_2$) by interpolating these with satellite proxy data. Access to in-situ $pCO_2$ data, via platforms such as SOCAT (Surface Ocean $CO_2$ Atlas), has been crucial to the success of empirical methods (Rödenbeck et al., 2015; Bakker et al., 2016). This, in conjunction with the increasing use of machine learning, has seen a proliferation in the number and diversity of methods in the literature. Rödenbeck et al. (2015) compared a suite of fourteen methods using a regional framework provided by Fay and McKinley (2014). The majority of these methods are variants of multiple linear regression (MLR) or artificial neural networks (ANN), with regression being applied in regional windows or clusters based on climatologies of satellite measurable variables. The authors found that methods agreed in regions where data coverage was adequate, but for data sparse regions, such as the Southern Ocean, interannual $CO_2$ variability of various empirical methods were not coherent.

Only two of the methods in Rödenbeck et al. (2015) were able to adequately represent interannual variability of $\Delta pCO_2$, namely: the SOM-FFN (self-organizing map – feed forward neural network) from Landschützer et al. (2014), and the mixed layer scheme (MLS) from Rödenbeck et al. (2014). These two methods were used by Landschützer et al. (2015) to show that Southern Ocean $CO_2$ uptake strengthened after 2000. However, these methods often showed large interannual differences in flux estimates despite agreeing on the overall decadal trend. This shows that there is lack of coherence even amongst the methods that perform well, meaning that different methods may lead to different interpretation of the drivers of $\Delta pCO_2$. The primary reason for the varied results is thought to be the way in which the algorithms deal with sparse data in the Southern Ocean (Rödenbeck et al., 2015). This alludes to the importance of testing multiple approaches, as different methods may be able to better represent the $CO_2$ estimates in the data sparse Southern Ocean.

In this study we introduce two methods new to this application, namely: Support Vector Regression (SVR) and Random Forest Regression (RFR). SVR is a method based on the theory of statistical learning, making the method robust to over-fitting by statistically determining the complexity of a problem rather than a heuristic approach as required in setting up an ANNs hidden layer structure (Vapnik, 1999; Smola et al., 2004). In a review on the use of Support Vector Machines (the broad category for regression and classification variants) in remote sensing, (Mountrakis et al., 2011) found that the method had the "ability to generalize well even with limited training samples". This makes SVR an appealing consideration for the sparsely sampled Southern Ocean. RFR uses an ensemble of decision trees to create robust estimates, often without requiring data preprocessing making it an effective "off the shelf" method (Louppe, 2014). As with SVM, Random Forests (both classification

and regression variants) have also been used in remote sensing applications, though it does not seem to be as widely used in earth systems sciences despite proving to be a powerful, yet easy to implement, learning algorithm (Caruana and Niculescu-Mizil, 2006; Hastie et al., 2009). We use SVR and RFR to estimate $CO_2$ fluxes in the Southern Ocean to try to better resolve the seasonal cycle from 1998 to 2014. These methods are trained with SOCAT v3 data collocated with satellite proxies. We compare these results with those of Landschützer et al. (2014). However, the lack of data in the Southern Ocean, particularly in winter, makes it difficult to understand the limitations of these methods within the context of SOCAT data.

To gain a better understanding of these methods' strengths and weaknesses we implement SVR and RFR in a synthetic data environment. A similar approach was taken by Friedrich and Oschlies (2009) in the North Atlantic, which experienced a similar data paucity to the Southern Ocean in the early 2000's. This idealised environment was also used to estimate the effect of including/excluding certain proxy variables as well as the optimal coverage of cruise tracks to constrain the North Atlantic $\Delta p CO_2$ adequately. Similarly, we assess the efficacy of including coordinate variables as proxies of $\Delta p CO_2$ in the empirical methods. In the intercomparison study by Rödenbeck et al. (2015) proxies typically include, but are not limited to sea surface temperature (SST), chlorophyll-a (Chl-$a$), mixed layer depth (MLD) and sea surface salinity (SSS); however several methods in the study also include latitude and longitude. While coordinates do not mechanistically impact $\Delta p CO_2$, they do help to constrain estimates where the available remote sensing proxies cannot adequately do so. The synthetic data is also used to test the ability of the SVR and RFR to approximate $\Delta p CO_2$ in the seasonally sparse Southern Ocean.

## 2  Data and Methods

This study is presented in two parts. The first applies SVR and RFR to the SOCAT v3 dataset and compares these outputs with those of the SOM-FFN by Landschützer et al. (2014). These estimates will be referred to as the observational estimates. Here the domain is limited to the three Southern Ocean (SO) domains of Fay and McKinley (2014) that are shown in Figure 1. These biomes are used to assess the performance of each of the methods, as done in Rödenbeck et al. (2015). Fay and McKinley (2014) use a different nomenclature, which roughly corresponds to frontal zones. We rename the Sub-Tropical Seasonally Stratified biome (STSS) as the Sub-Antarctic Zone (SAZ); the Sub-Polar Seasonally Stratified biome (SPSS) becomes the Polar Frontal Zone (PFZ) and the ice biome (ICE) is the Antarctic Zone (AZ) (Mongwe et al., 2016).

The second part aims to better understand the limitations of these methods with the given dataset by implementing the methods to ocean biogeochemical model output. The domain of this synthetic data experiments is defined by the three southern biomes of Fay and McKinley (2014). These are defined by observed oceanographic and biological parameters, but are used for the sake of consistency despite potential differences between observations and the model.

### 2.1  Gridded Data

The data sources are shown in Table 1. These gridded data refer primarily to remotely sensed data, with the exception of MLD and SSS. The latter variables are output from $ECCO_2$, an assimilative model. The temporal range of the data (1998 through 2014) is limited by the availability of Globcolour (Chl-$a$ starting in 1998) and SOCAT v3 ($f CO_2$ ending in 2014).

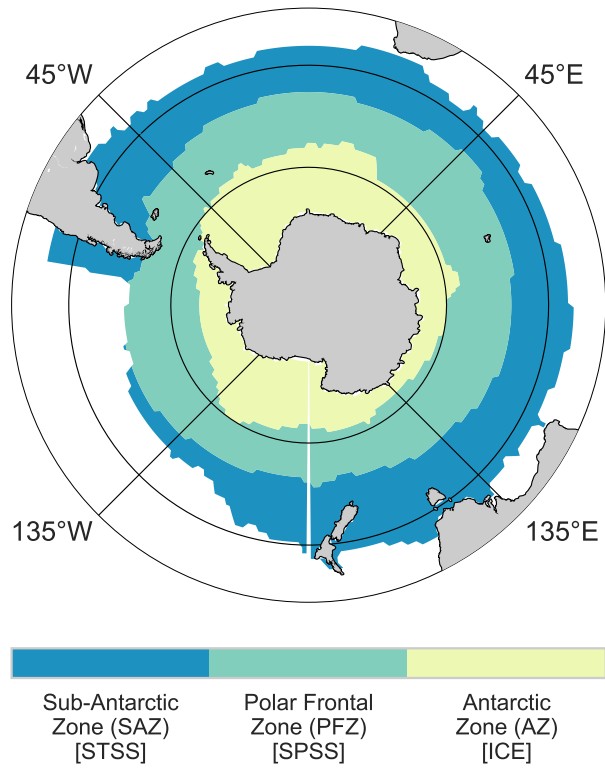

**Figure 1.** The three Southern Ocean biomes as defined by Fay and McKinley (2014). The common names for the biomes are shown in the key, with the abbreviations shown in the round brackets. The abbreviation in the square brackets show the abbreviations as given by Fay and McKinley (2014).

**Table 1.** Information on data products used in this study. The temporal and spatial resolutions are for the raw data (before gridding). Dashes show that times are either not applicable or that the dataset is continually updated. Note that the start and end year show full years only. Links to download the data are given in the additional materials. The asterisk (*) indicates that variables are the output of a data assimilative model.

| Group / Product | Variables | Date Range | | Resolution | | Reference |
| --- | --- | --- | --- | --- | --- | --- |
| | | Start | End | Time | Space | |
| SOCAT v3 | $f\mathrm{CO}_2^{sea}$ | 1970 | 2014 | 1 mon | 1° | (Bakker et al., 2016) |
| CDIAC | $x\mathrm{CO2}_2^{atm}$ | 1970 | 2014 | – | – | (Masarie et al., 2014) |
| Globcolour | Chlorophyll | 1998 | – | 1 day | 0.25° | (Maritorena and Siegel, 2005) |
| GHRSST | Sea Surface Temperature | 1981 | – | 1 day | 0.25° | (Reynolds et al., 2007) |
| ECCO2 (cube92) | *Mixed Layer Depth *Salinity | 1992 | 2015 | 1 day | 0.25° | (Menemenlis et al., 2008) |

All data are gridded to monthly x 1° using *iris* and *xarray* packages in Python (Hoyer and Hamman, 2017; Met Office). Gridded $pCO_2$ (SOCAT v3) is used to train the algorithms (Bakker et al., 2016). Surface station measurements (flask and tower) of atmospheric $xCO_2$ are interpolated to a regular grid using support vector regression (Masarie et al., 2014). Mean sea level pressure (NCEP2) is used in the conversion from $xCO_2$ to $pCO_2$ (Kanamitsu et al., 2002).

Cloud coverage and low light at high latitudes during winter result in missing Chl-$a$ data. Cloud gaps are filled with the climatology of Chl-$a$ (from 1998 to 2014) and missing low light data are filled with a value of $0.1 \pm 0.03$ mg m$^{-3}$ (uniformly distributed random noise).

## 2.2   Model Data

The output from a regional NEMO-PISCES configuration (BIOPERIANT05-GAA95b) is used as the synthetic dataset. The
configuration is an updated version of PERIANT05 used by Dufour et al. (2012), where BIOPERIANT05-GAA95b includes biogeochemistry with PISCES-v2. The model has a peri-Antarctic domain with an open northern boundary at 30°S. The horizontal resolution of the configuration is 0.5° cos(latitude) with 46 vertical levels. The northern boundary is forced by a global 0.5° model, ORCA05 as presented in Biastoch et al. (2008). Output is saved as five-day averages. The simulation was run from 1998 to 2009. The synthetic observations are sampled at the model resolution (5-day × 0.5°) to resemble the SOCAT
dataset. Hereafter all data is resampled to 1.0° spatial resolution and monthly temporal resolution data to match observations. Finally, for the simulation experiment we define the Southern Ocean using the three southernmost biomes defined in Fay and McKinley (2014) as done for the observational estimates.

## 2.3   Data transformation and derived variables

Both gridded data and synthetic input data are transformed in preparation for the empirical algorithms. The $\log_{10}$ transforma-
tions of MLD and filled chlorophyll (Chl-$a_{clim}$) are taken to return a distribution that closer represents a normal distribution.

    Several of the studies in Rödenbeck et al. (2015) included latitude, longitude and/or time as proxies of $\Delta pCO_2$. It is important to note that coordinates do not drive mechanistic changes in $\Delta pCO_2$. Rather, the inclusion of coordinates in the empirical methods account for unknown or regionally varying proxies that cannot be measured remotely. Many methods in the intercomparison by Rödenbeck et al. (2015) did not include coordinates, but account for unaccountable spatial variability by clustering
or subsetting data regionally. In this study, we use a single large domain with no clustering or regional subsets. Two scenarios for each method in the simulation experiment are run: no coordinate variables, and including coordinate variables (time, latitude and longitude).

    The coordinates are transformed to preserve the continuity of the data as is shown below. Seasonality of the data is preserved by transforming the day of the year ($j$) and is included in both SVR and RFR analyses:

$$t = \begin{pmatrix} \cos\left(j \cdot \frac{2\pi}{365}\right) \\ \sin\left(j \cdot \frac{2\pi}{365}\right) \end{pmatrix} \tag{1}$$

Transformed coordinate vectors were passed to both SVR and RFR using n-vector transformations of latitude ($\lambda$) and longitude ($\mu$) (Gade, 2010; Sasse et al., 2013), with n containing:

$$A, B, C = \begin{pmatrix} \sin(\lambda) \\ \sin(\mu) \cdot \cos(\lambda) \\ -\cos(\mu) \cdot \cos(\lambda) \end{pmatrix} \tag{2}$$

Co-located fCO$_2$ (y) and proxy data (X) are used to create training arrays (x). The final input for SVR and RFR are (with 12 columns): $\log_{10}$(Chl-$a_{clim}$), SST, $f$CO$_{2(atm)}$, ADT, $\log_{10}$(MLD), ICE, SSS, $\cos(j)$, $\sin(j)$ and n-vectors [A, B, C]. SVR requires each column of the proxies to be z-scored; *i.e.* normalized to the mean ($\mu$) and standard deviation ($\sigma$) of each column $\left(\frac{x-\mu}{\sigma}\right)$.

## 2.4 Empirical methods and implementation

Data are split randomly into a training and independent test dataset with a ratio $0.7 : 0.3$. The independent dataset is used to give a test error of the trained algorithm. The statistical learning package, *Scikit-Learn*, in Python is used for all regression and cross-validation methods (Pedregosa et al., 2011). The details on each cross-validation method are outlined in the subsections below.

### 2.4.1 Support vector regression

The basic formulation of SVR is similar to that of linear regression as described by Smola et al. (2004):

$$f(x) = \langle w, \mathbf{x} \rangle + b \qquad \text{with } b \in \mathbb{R} \tag{3}$$

where $b$ is an intercept, $\langle \cdot, \cdot \rangle$ denotes the dot product of the weights ($w$) and $\mathbf{x}$, the training data. The weights and intercept are found by solving the cost function:

$$\text{minimise} \quad \frac{1}{2}||w||^2 \text{ subject to} \qquad \begin{cases} y_i - \langle w, x_i \rangle - b & \leq \epsilon \\ \langle w, x_i \rangle + b - y_i & \leq \epsilon \end{cases} \tag{4}$$

In this form, $w$ is minimised according to the target values ($y_i$) to a precision of $\epsilon$ – *i.e.* there is no room for error greater than $\epsilon$. However, with the majority of problems, meeting these constraints is not possible if data are noisy or $\epsilon$ is set small. The inclusion of *slack variables* ($\xi_i, \xi_i^*$) relaxes the constraints and the problem is now formulated as:

$$\text{minimise} \quad \frac{1}{2}||w||^2 + C\sum_{i=1}^{n}(\xi_i + \xi_i^*) \qquad \text{subject to} \quad \begin{cases} y_i - \langle w, x_i \rangle - b & \leq \epsilon + \xi_i \\ \langle w, x_i \rangle + b - y_i & \leq \epsilon + \xi_i^* \\ \xi_i, \xi_i^* & \geq 0 \end{cases} \tag{5}$$

Here $C$ is a parameter that adjusts for the amount of error that the minimisation allows. The slack variable $|\xi|$ is only counted towards the cost if the point lies outside the margin ($|\xi| \geq \epsilon$). The points on or outside the margins are called support vectors

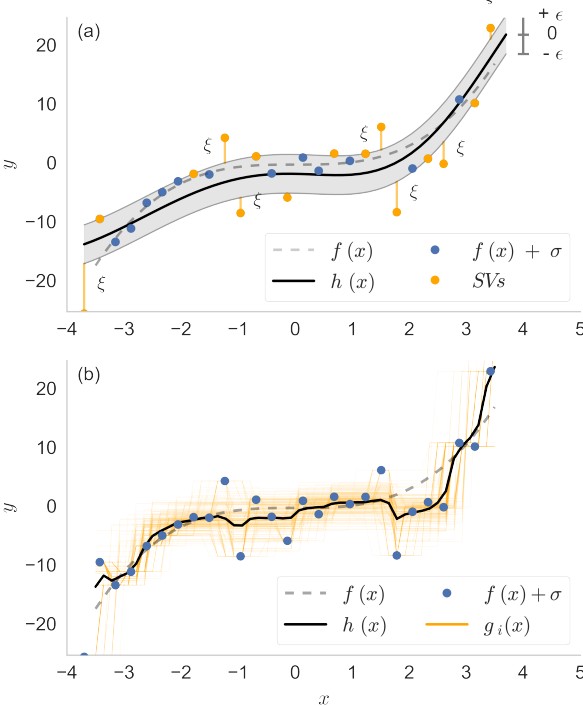

**Figure 2.** A simple example demonstrating the principle of (a) support vector regression and (b) random forest regression. The dashed grey line is the true function $f(x) = 0.4x^3$ with the blue dots representing a random sample taken from this function $f(x)+\sigma$, where $\sigma$ is normally distributed noise. The black line in each figure, $h(x)$, show the estimate of the true function. The orange dots in (a) show the samples from the random subset chosen as support vectors from which $h(x)$ is estimated. The orange lines in (b) show 200 decision tree estimates, $g_i(x)$, which are averaged to create the ensemble, $h(x)$.

and are used to construct the hypothesis function, $h(x)$. This is shown in Figure 2a where a linear SVR is fitted to noisy data produced from a cubic spline. The optimisation problem shown in Eq. 5 is solved in its dual formulation (see Hastie et al. 2009 for the full description). Importantly, solving the dual formulation allows for efficient kernelisation of SVR.

Kernelisation describes the process that maps the proxy variables ($\mathbf{x}$) onto a higher dimensional feature space. In this study we used a Gaussian kernel (or radial basis function – RBF), which allows for potentially infinite complexity determined by the number of support vectors (Vapnik, 1999). The RBF kernel introduces an additional hyper-parameter ($\gamma$) that defines the width of the Gaussian. Selection of the SVR hyper-parameters ($\epsilon$, $C$, $\gamma$) is done using a two-stage exhaustive grid search approach with cross validation. We use K-fold cross validation, where the data is divided into eight equal "folds" ($k = 8$). Seven of the folds are used to train the model, while the remaining fold is used for validation. This is done iteratively until each excluded fold has been used to test the results.

### 2.4.2 Random Forest Regression

Decision trees form the basic building block of a Random Forest (RF), with the average of $n$ decision tress is taken as the ensemble estimate (Breiman, 2001) (Figure 2b). The basic idea of a decision tree is to iteratively partition data into boxes using simple rules that minimize the error at each split (referred to as a node) – these boxes would become hypercubes in higher dimensional problems. This is described by the basic formulation as described in Loh (2011):

1. *Start at the root node*

2. *For each X, find the set S that minimizes the sum of the node impurities in the two child nodes and choose the split $X \in S$ that gives the minimum overall X and S.*

3. *If a stopping criterion is reached, exit. Otherwise, apply step 2 to each child node in turn.*

Decision trees have high variance due to their discrete nature. Random forests reduce this high variance by bootstrapping with aggregation (called *bagging*): a subset of the available training dataset is sampled with replacement for each decision tree in the RF. The sampling with replacement means that each training observation has a $\sim 63\%$ chance of being chosen at least once for a particular tree (Louppe, 2014). This subsampling provides estimates that are robust to outliers as these have a chance of being omitted in training. This means that a random forest typically performs better when number of decision trees ($t$) is large, but increasing the number of trees has diminishing returns in terms of performance vs. computation. Additional robustness is given to RFs by randomizing and/or limiting the number of proxy variables ($m$) given to the nodes in each tree when splitting the data (hence random) (Louppe, 2014). In this study, the maximum number of proxy variables ($m = 11$) was given to the RFR. The complexity of a RF can be adjusted by limiting the minimum number of leaves at a terminal branch ($l$), where a fully-grown tree would allow $l$ to be one.

A useful feature of bagging is that it intrinsically provides a cross-validation dataset (a.k.a. out-of-bag samples) that is not part of the training procedure (for a specific set of trees). The out-of-bag samples are those that are not selected during bagging. The advantage of this approach over K-fold cross-validation is that the full dataset can be used in the training procedure, as opposed to splitting the dataset for cross-validation. The out-of-bag error is used to cross-validate the model and select the hyper-parameters ($t$, $m$, $l$) for the RF.

### 2.5 $CO_2$ fluxes

Air-sea $CO_2$ fluxes are quantified with:

$$FCO_2 = K_0 \cdot k_w \cdot \Delta pCO_2 \cdot (1 - [\text{ice}]) \tag{6}$$

The gas transfer velocity ($k_w$) is calculated using a quadratic dependency of wind speed with the coefficients of Wanninkhof (2014). The $u$ and $v$ vectors of CCMP v2 are used to compute the wind speed (Atlas et al., 2011). Coefficients from Weiss (1974) are used for $K_0$ and $\Delta pCO_2$ is estimated by the empirical models. The effect of sea-ice cover on $CO_2$ fluxes is treated linearly; the fraction of sea ice cover ([ice]) is converted to fraction of open water by subtracting one as shown in Equation (6).

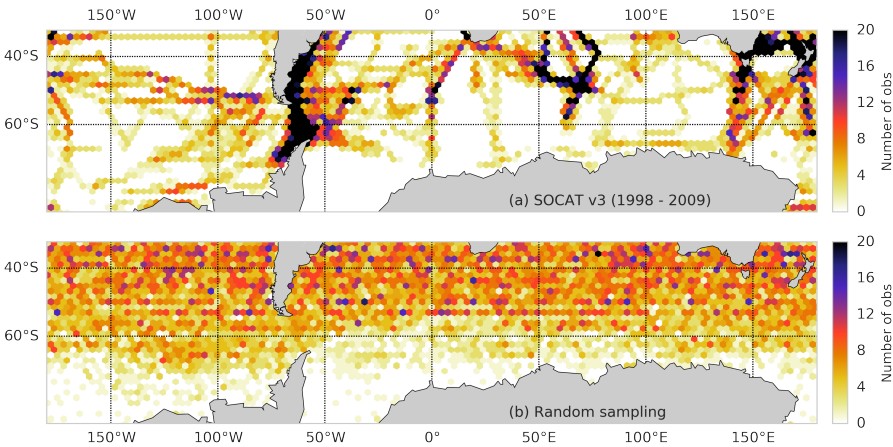

**Figure 3.** The spatial distribution of sampling locations in the synthetic dataset (BIOPERIANT05). The top panel (a) shows the sampling strategy using SOCAT v3 locations; and the bottom panel (b) shows the uniform random sampling distribution used in the second experiment.

These results are analyzed regionally with the three Southern Ocean biomes defined by Fay and McKinley (2014) (Figure 1). We compare our estimates of $CO_2$ fluxes with those of Landschützer et al. (2014) who used a two-step neural network method abbreviated to SOM-FFN (self-organizing map – feed forward neural network). Note that the SOM-FFN method was trained using SOCAT v2 compared to the methods in this study that used SOCAT v3.

## 2.6 Synthetic data experiments

Two experiments are run with the synthetic data. The first experiment aims to identify the efficacy of including or omitting coordinates as proxy variables on each method's ability to estimate $\Delta p CO_2$ using SOCAT v3 locations. This is achieved by implementing the model with the transformed coordinate variables as proxies and then without. Note that the training procedure for the models remains the same as for the observational estimates of $\Delta p CO_2$.

The second experiment assesses the impact that the seasonally sparse SOCAT v3 has on the ability of the methods to estimate $\Delta p CO_2$. This is done by comparing the results of $\Delta p CO_2$ estimates when trained according to: 1) SOCAT v3 locations trained with synthetic data (Figure 3a); 2) uniformly random sampling locations (random in space and time) with a sample size the same as SOCAT v3 (Figure 3b). Once again this the training procedure remains the same (as stated above).

## 3 Results

## 3.1 Observational $CO_2$ data results

We use the root mean squared error (RMSE) as the primary metric of the methods' performance as shown in Figure 4a–c. Note that the RFR RMSE is calculated from the out-of-bag error (effectively an independent error). SOM-FFN has the best RMSE

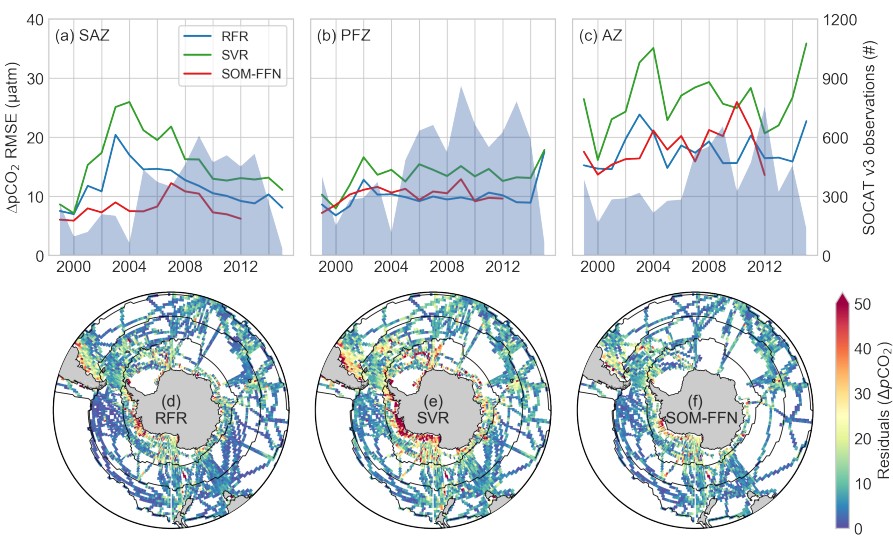

**Figure 4.** The RMSE (top row, a–c) for each of the three Southern Ocean biomes for RFR (blue), SVR (green) and SOM-FFN (red). The grey fill in the top row (a–c) shows the number of observations for each of the biomes for each year. The maps in the bottom row (d–f) show the spatial distribution of residuals in the Southern Ocean for SVR (d), SOM-FFN (e) and RFR (out-of-bag errors) (f). The thin black lines define the three Southern Ocean biomes as defined by Fay and McKinley (2014). Note that RFR and SVR are trained and tested with SOCAT v3 while SOM-FFN is trained and tested with SOCAT v2.

score of 14.84 µatm (using SOCAT v2), which is better than the RMSE of RFR (16.45 µatm) and SVR (24.404 µatm), which are trained with SOCAT v3. The biases of the different methods are similar in magnitude for each of the biomes (-0.40, -0.03 and -0.75 µatm for the SOM-FFN, RFR and SVR respectively). The mean absolute errors (MAE) for the respective methods are 9.78, 9.85 and 15.27 µatm respectively.

The difference between the mean absolute error (MAE) and the RMSE informs on the ability of methods to fit outliers or extreme points, as the RMSE scores larger errors much more severely than MAE. The SOM-FFN approach has the smallest difference between these two metrics (5.06, 6.60 and 9.13 µatm for the SOM-FFN, RFR and SVR respectively). This superior performance may be due to two factors. Firstly, the SOM-FFN method may be better at fitting the extreme points (those that are in the outer percentiles of the distribution). Second, it may allude to the fact that the SOCAT v2 dataset is less variable.

Testing the SVR and RFR implementations against SOCAT v2 yields similar results, with the exception of in the SAZ, where both RMSE and MAE improve (results shown in Table 2).

    The RMSEs and biases in the PFZ are least variable between methods. While there is a substantial increase in the number of observations from 2004 there is no appreciable change in the RMSE. The Antarctic Zone (AZ) is the primary contributor to these errors with much larger average RMSE values than for the SAZ and PFZ (36.14, 23.80 and 21.32 µatm for SVR, RFR

and SOM-FFN respectively). This increase in the RMSE is likely driven by the larger variability of $\Delta p\text{CO}_2$ observations in the

**Table 2.** Various performance metrics for empirical estimates of $\Delta p\mathrm{CO}_2$ in the Subantarctic Zone (SAZ), Polar Frontal Zone (PFZ) and Antarctic Zone (AZ) (as defined by Fay and McKinley 2014). Results tested according to SOCAT v2 and SOCAT v3 are shown for the SVR and RFR methods.

|  | Method | RMSE | MAE | $r^2$ | Bias |
|---|---|---|---|---|---|
|  | SVR (v3) | 18.14 | 11.28 | 0.48 | 0.61 |
|  | SVR (v2) | 15.99 | 10.36 | 0.49 | 0.20 |
| SAZ | RFR (v3) | 13.67 | 8.16 | 0.70 | -0.14 |
|  | RFR (v2) | 12.66 | 7.65 | 0.68 | -0.29 |
|  | SOM-FFN | 10.07 | 7.04 | 0.76 | -1.15 |
|  | SVR (v3) | 14.45 | 10.06 | 0.48 | 0.31 |
|  | SVR (v2) | 14.29 | 10.01 | 0.44 | -0.01 |
| PFZ | RFR (v3) | 10.71 | 6.7 | 0.71 | 0.21 |
|  | RFR (v2) | 10.56 | 6.77 | 0.69 | -0.34 |
|  | SOM-FFN | 11.01 | 7.68 | 0.6 | 0.26 |
|  | SVR (v3) | 36.14 | 25.19 | 0.56 | -3.22 |
|  | SVR (v2) | 35.69 | 25.01 | 0.59 | -2.88 |
| AZ | RFR (v3) | 23.8 | 15.81 | 0.8 | -0.27 |
|  | RFR (v2) | 23.49 | 15.63 | 0.81 | -0.62 |
|  | SOM-FFN | 21.32 | 14.91 | 0.82 | -0.77 |

AZ, where standard deviations of observations are 25.05, 20.01 and 54.65 µatm for the SAZ, PFZ and AZ respectively. This is reflected in the highest $r_2$ scores in the AZ for the respective methods (Table 2).

The annual and seasonal averages (winter = JJA, summer = DJF) for $\Delta p\mathrm{CO}_2$ estimated by RFR, SVR and SOM-FFN for the Southern Ocean are shown in Figure 5. Note that the estimates have been scaled to sea ice concentration ($\Delta p\mathrm{CO}_2 \times (1 - [ice])$)
as done for fluxes in Equation 6 – this mutes winter estimates in the AZ. There is, in general, good agreement in the spatial distribution between the methods with the SAZ being a net sink of $\mathrm{CO}_2$ and the region south of the Polar Front (PFZ and AZ) a source of $\mathrm{CO}_2$ to the atmosphere as found by Metzl et al. (2006).

More specifically, there is stronger zonal asymmetry in summer compared to winter. This is driven, in part, by a strong reduction of $p\mathrm{CO}_2$ by biological production (Metzl et al., 2006; Lenton et al., 2012). There are three regions in the SAZ where
$\Delta p\mathrm{CO}_2$ reduction is strongest and consistent between methods (Figure 5): east of South America (Malvinas Confluence), south east of Africa (Agulhas retroflection) and between Australia and New Zealand (Tasman Sea). The reduction of $\Delta p\mathrm{CO}_2$ in the PFZ is strongest in the Atlantic sector downstream of the South Sandwich and South Georgia Islands and in the Indian sector downstream of the Kerguelen Plateau (Figure 5d-f). In both cases, SAZ and PFZ, these regions are consistent with regions of high biomass (Thomalla et al., 2011; Carranza and Gille, 2015).
However, there are more subtle differences in the magnitudes and distributions of these patterns. The RFR underestimates winter outgassing south of the Polar Front (Figure 5g) compared to the other methods resulting in a weaker annual source.

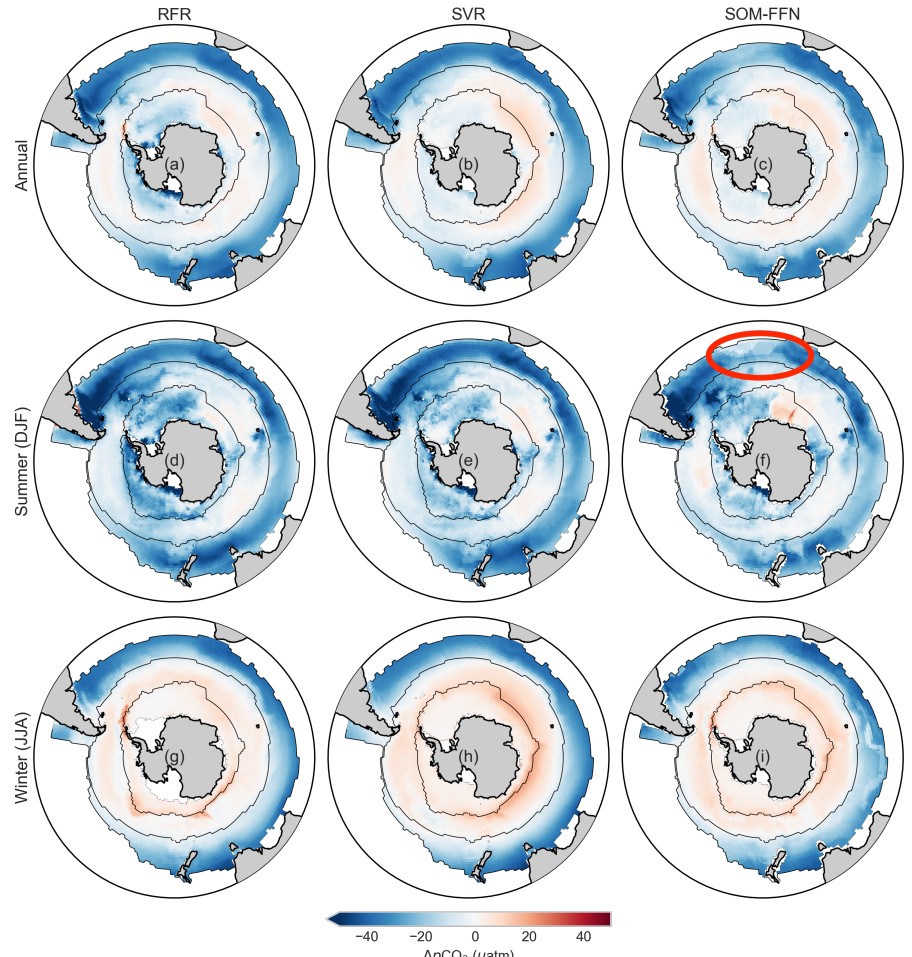

**Figure 5.** Seasonal averages for $\Delta p\mathrm{CO}_2$ from 1998 to 2014 for SVR, SOM-FFN and FRF. The annual mean is shown in the top row (a, b, c); the mean summer (DJF) $\Delta p\mathrm{CO}_2$ is shown in the middle row (d, e, f); and the mean winter (JJA) $\Delta p\mathrm{CO}_2$ is shown in the middle row (g, h, i). The thin black lines denote the SAZ, PFZ and AZ from outside inward. Note that the $\Delta p\mathrm{CO}_2$ has been normalized to sea ice cover where $\Delta p\mathrm{CO}_2$ is multiplied by $(1 - [\mathrm{ice}])$. The red oval in (e) highlights the difference in SOM-FFN estimates of $\Delta p\mathrm{CO}_2$ during summer in the Atlantic compared to SVR and RFR.

Conversely, the SVR has strong winter outgassing (Figure 5h) in the PFZ compared to other methods. In summer, the largest difference occurs in the eastern Atlantic sector of the SAZ where the SOM-FFN $\Delta p\mathrm{CO}_2$ estimates (highlighted in Figure 5f) are larger compared to SVR and RFR. Other differences in the spatial output are more subtle.

The agreements and differences between methods are also observed in the time series for each of the biomes (Figure 6). Importantly there is coherence in the strengthening sink (2002 to 2012) and timing of the seasonal cycle between the three methods (Landschützer et al., 2015). The differences in the magnitude of the winter outgassing in the PFZ and AZ (Figure

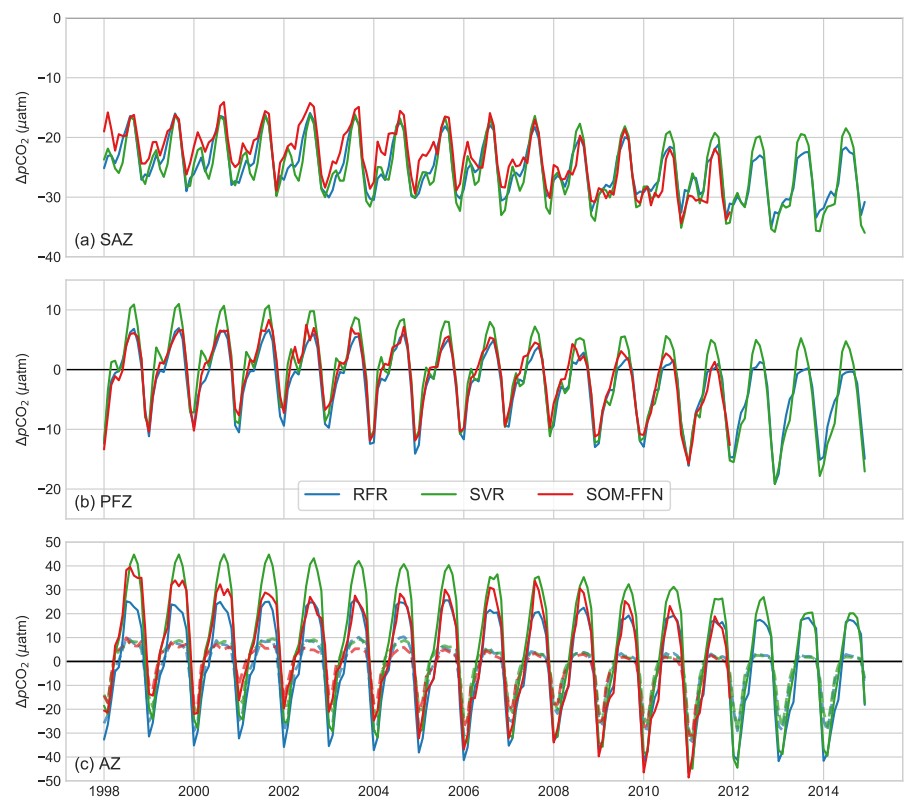

**Figure 6.** Time-series of $\Delta p\mathrm{CO}_2$ estimates for the three Southern Ocean biomes as defined by Fay and McKinley (2014): SAZ, PFZ and MIZ. The y-axis gridlines represent the same scale for figures (a) through (c). The SOM-FFN estimates are only available until 2011 as it is trained with SOCAT v2, while the SVR and RFR are trained with SOCAT v3. $\Delta p\mathrm{CO}_2$ normalised to sea ice cover is shown by dashed lines in the AZ.

6b,c) are also apparent, with the SVR overestimating $\Delta p\mathrm{CO}_2$ compared to other methods and the RFR with conservative outgassing estimates.

There is also a large difference between the SOM-FFN and the two other methods in summer, particularly from 1998 through 2006. Figure 5f shows that this could be driven by the difference in the eastern sector of the Atlantic (circled with red). Estimates of winter $\Delta p\mathrm{CO}_2$ are in agreement, with the exception of the last four years when SVR winter estimates increase relative to RFR. The SAZ and PFZ also show variability in the magnitude of a seasonal shoulder in late summer, where increasing $\Delta p\mathrm{CO}_2$ is briefly delayed by a short sharp decrease resulting in a saw-tooth pattern. This effect is the strongest for the SVR and weakest for the RFR.

The seasonal amplitude of $\Delta p\mathrm{CO}_2$ in the AZ is far larger than in the SAZ and PFZ (Figure 6c) resulting in large methodological differences. However, this large differential is not realized in calculated air-sea $\mathrm{CO}_2$ fluxes, due to ice cover as shown

**Table 3.** Root mean squared error (RMSE - µatm) for the three synthetic data experiments for RFR (left), SVR (middle) and the ensemble mean (ENS) of the two methods. Both in- and out-of-sample errors are reported (E[in] and E[out] respectively). SOCAT experiments are those where the location of synthetic training data is the same as SOCAT v3. This was run with (W coords) and without coordinates (W/O coords) namely time, latitude and longitude. A third experiment was run with random samples – coordinate variables are included as proxies.

|        | Experiment          | RFR  | SVR  | ENS  |
|--------|---------------------|------|------|------|
|        | SOCAT (W/O coords)  | 6.65 | 7.47 | —    |
| E[in]  | SOCAT (W coords)    | 5.12 | 5.10 | —    |
|        | Random Sampling     | 7.23 | 7.83 | —    |
|        | SOCAT (W/O coords)  | 7.46 | 7.46 | 7.08 |
| E[out] | SOCAT (W coords)    | 5.76 | 6.19 | 5.36 |
|        | Random Sampling     | 4.88 | 4.94 | —    |

by the dashed lines (Figure 6c). Summer estimates are also influenced by sea ice cover, but not to the extent that winter fluxes would be reduced.

## 3.2 Simulation experiment results

The advantage of using synthetic data is that both in- and out-of-sample errors can be estimated, where the in-sample error is
calculated from the training points and the out-of-sample error from the entire predicted domain. The latter gives a representation of the *true* error of the method. The results from these experiments are shown in Table 3. The detailed out-of-sample histograms are shown in Figure B1.

### 3.2.1 Coordinates as proxy variables

This experiment used the synthetic dataset to test the influence of including or excluding transformed coordinates (time, latitude
and longitude) as proxies of $\Delta p\text{CO}_2$. There are four major results from the experiment results. Firstly, the RMSE estimates are smaller when coordinates are included as proxies for both in- and out-of-sample subsets (Table 3). Secondly, RFR achieves marginally better out-of-sample RMSE than SVR (5.76 and 6.19 µatm respectively) when trained with coordinates. Third, both RFR and SVR have comparable out-of-sample RMSE estimates (7.46 µatm) for $\Delta p\text{CO}_2$ estimates trained with and without coordinate proxies. Lastly, the ensemble mean of SVR and RFR has lower out-of-sample RMSE estimates than the individual
estimates for implementations with and without coordinate proxies, though these gains are marginal (Table 3).

These points can also be gleaned from RMSE maps (Figure 7). Both RFR and SVR errors are low; however the RFR outperforms the SVR marginally for the open ocean regions. Errors in coastal regions remain high for each of the experiments and methods (Figure 7a,b,d,e); such as in the Argentine Sea, the Agulhas retroflection, and the marginal ice zone. The ensemble mean of the estimates achieves a balance between the two methods with low and moderate RMSE scores in the open ocean
and coastal regions. Lastly, the distributions of the errors for RFR and SVR without coordinate proxies (and thus the ensemble mean) are similar.

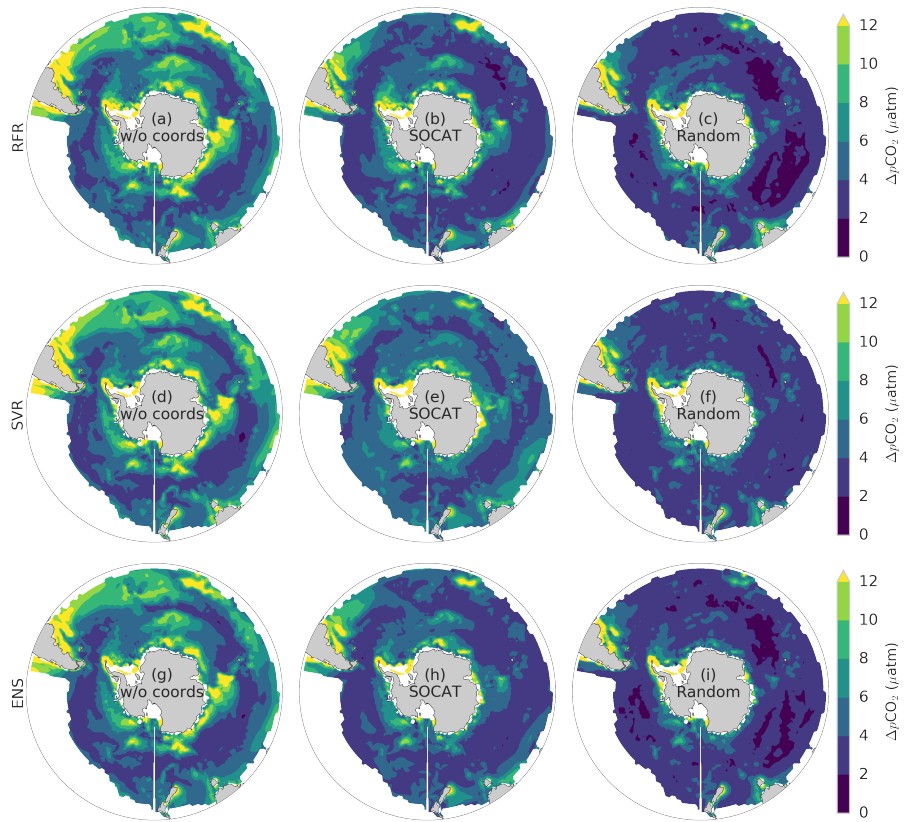

**Figure 7.** Distributions of root mean squared error (RMSE) for the three synthetic data experiments for RFR in the top row (a–c), SVR in the middle row (d–f) and the ensemble mean for RFR and SVR in the bottom row (g–i). The first column (a, d, g) shows the RMSE for synthetic SOCAT training locations without coordinates as proxies, while the second column (b, e, h) includes coordinates as proxies. The last column (c, f, i) shows the RMSE of randomly sampled training locations where coordinates are included as proxies.

The time series (Figure 8) show that including coordinate variables plays an important role in achieving accurate phasing of the seasonal cycle. When coordinates are not included as proxies the phasing shifts earlier for both methods. There is also an improvement of estimates over time, where the first two years (1998 and 1999) have worse estimates for both SVR and RFR (Figure 8). This does not seem to be linked to the number of observations, but could be due to the distribution. The ensemble $\Delta p\text{CO}_2$ in the 1998 to 1999 period is closer to BIOPERIANT05 output as the respectively over- and underestimates of RFR and SVR compensate for each other.

### 3.2.2 Random sampling regime

This experiment is performed to assess the inaccuracies that arise due to the spatial and temporal sampling biases in the SOCAT v3 dataset. A random sampling regime is compared to the Training locations are chosen at random and uniformly in time and

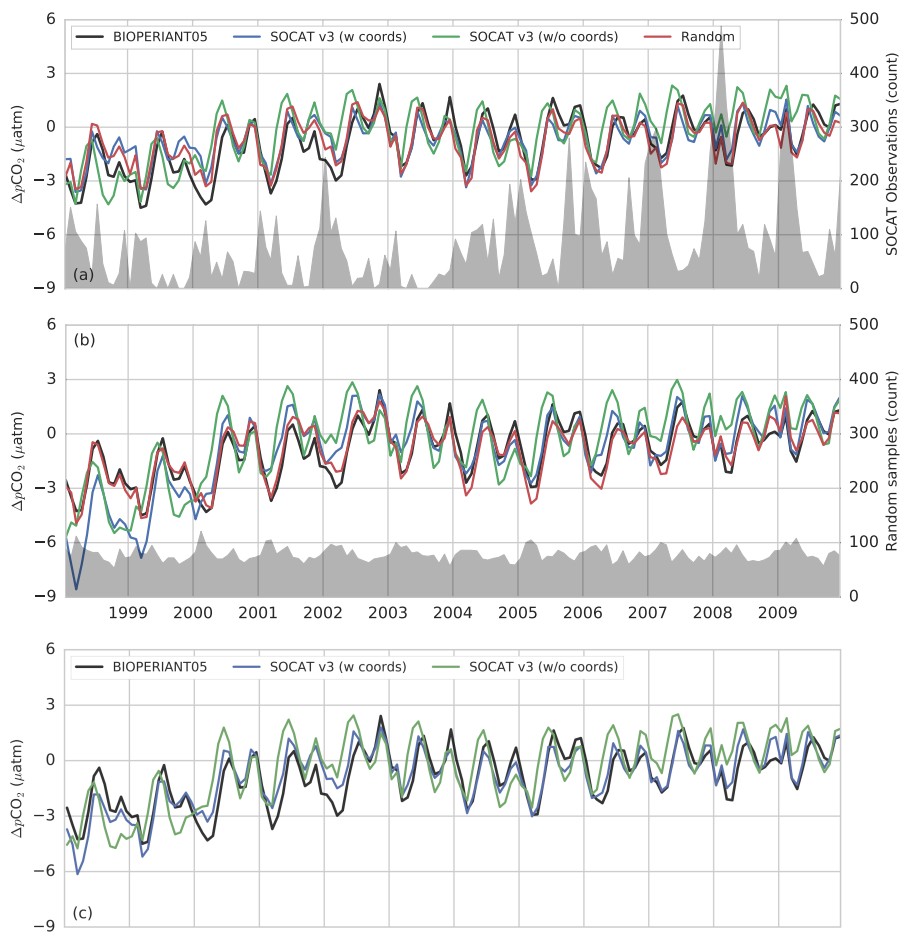

**Figure 8.** Time series of BIOPERIANT05 $\Delta p$CO$_2$ (target) and empirical estimates of $\Delta p$CO$_2$ for each of the experiments for RFR (a), SVR (b) and the ensemble mean estimates (c). The SOCAT v3 estimates are trained using the locations of SOCAT v3 data. The *w coords* variant includes coordinates as proxies of $\Delta p$CO$_2$ while these are not included for *w/o coords*. The *Random* estimates are trained with uniformly distributed random sample locations. The number of samples per time step for SOCAT (a) and random sampling locations (b) are shown by the grey fill.

space. This eliminates any summer/winter biases as well as clustering of cruise tracks in certain regions (such as the Argentine sea). Note that coordinates are included as proxies of $\Delta p$CO$_2$ with the random sampling regime.

Firstly, the results show that the biases in SOCAT v3 do contribute to out-of-sample errors, as the random sampling regime achieved lower RMSE scores than any of the other experiments (4.88 and 4.94 µatm for RFR and SVR respectively as in Table 3). However, RFR is marginally less susceptible to sampling biases than SVR as the relative improvement for the latter is larger (with differences of 0.88 and 1.70 µatm respectively). The spatial distributions of RMSE for the random sampling implementations (Figure 7c,f) show that errors in coastal regions remain high (> 12 µatm) with uniform sampling. Lastly,

there is an improvement in the estimates from 1998 to 2000 with the random sampling particularly for the SVR (Figure 8), suggesting that the method is more susceptible to the temporal bias than RFR (if coordinates are included as proxies).

## 4 Discussion

### 4.1 Observational estimates

In this section we address the methods' ability to fit the training data, in other words an assessment of in-sample errors (Figure 4 and Table 2). Thereafter we investigate the differences in the estimates of $\Delta p\mathrm{CO}_2$ (Figures 5 and 6).

#### 4.1.1 Assessment of in-sample errors

Based on the results, the SOM-FFN method (by Landschützer et al. (2014)) proves to be an elegant implementation of neural network methods that is able to estimate SOCAT $\Delta p\mathrm{CO}_2$ (in-sample estimate) better than the RFR and SVR methods (with respective RMSE estimates of 14.84, 16.45 and 24.40 µatm). Here we assess these differences and try to identify the possible reasons for the differences.

One of the largest differences in the methods ability to fit the training data is in the SAZ where the RFR and SVR score poorly in comparison to SOM-FFN, particularly from 2000 to 2006 (Figure 4a). This is during a period where the number of observations are still relatively low in the SOCAT v3 database (Figure 4a). This may then be due to an increase in the complexity of $\Delta p\mathrm{CO}_2$ estimates in the SAZ from SOCAT v2 to v3 from 1998 through 2006, thus more challenging to fit accurately. This is exemplified in the maps of RMSE (Figure 4g-i), where coastal regions typically have larger error estimates. A comparison of SOCAT v2 and v3 for this period shows that the increase in the number of observations occurs primarily in the Argentine Sea, thus confirming this hypothesis (Figure A1a). The comparison of SOCAT v2 and v3 RMSE results for RFR and SVR confirm this (Table 2), where there is a marked improvement when using the older dataset. Importantly, this shows that increasing the number of measurements does not necessarily improve the in-sample error estimates, but may yield a more accurate out-of-sample estimate; however this is difficult to test with limited data.

Despite the improvement in performance when testing against SOCAT v2, SVR and RFR still have poorer performance than the SOM-FFN approach. We attribute this in part, to the SOM-FFN's ability to reduce the large RMSE contributions observed in the other two methods. This notion is supported by the smaller difference between RMSE and MAE, especially in the SAZ (Table 2). The SOM-FFN achieves this by increasing the flexibility of the algorithm by having multiple regression models that can each be optimised for data with a particular length scale of variability. This allows the SOM-FFN approach to adapt to short scales of variability in dynamic regions such as the Argentine Sea and the coastal Antarctic (Figure 4i).

In comparison, this implementation of SVR, which is theoretically similar to an artificial neural network, only has one length scale for the entire domain (Vapnik, 1999; Smola et al., 2004). This becomes apparent in the AZ, where many of the observations are in the more biogeochemically dynamic coastal Antarctic, where melting sea ice results in short decorrelation length scales (Bakker et al., 2008; Chierici et al., 2012; Jones et al., 2012). The SVR has much larger RMSE scores in the AZ

than the RFR or SOM-FFN (35.69, 23.49 and 21.32 µatm respectively). This suggests that implementing the SVR approach without an initial clustering or regionalisation step, will not yield good results.

By comparison, the RFR approach is more adept at fitting various length scales of variability, accounting for both the higher and lower variability in the AZ and PFZ respectively (with SOCAT v3 standard deviations of 54.65 and 20.01 respectively).
The high $r^2$ scores achieved by RFR in the AZ and PFZ (0.81 and 0.71 respectively) highlight the flexibility in the method (Table 2). This is due to the differences in the underlying mathematics of the methods. Decision trees, which are the building block of RFR, separate data at each decision node with a discrete boundary (Breiman, 2001). Conversely, ANNs and SVRs often use Gaussian functions in the cost function, resulting in smoother approximations (Vapnik, 1999). This makes decision trees prone to overfitting, but the ensemble implementation of Random Forests eliminates this to a large extent.

### 4.1.2  Differences in $\Delta p\text{CO}_2$ estimates

One of the largest differences in $\Delta p\text{CO}_2$ is the weaker sink estimated by the SOM-FFN method in the SAZ (Figure 6). This difference can be traced to the eastern Atlantic SAZ, where the SOM-FFN has higher estimates of $\Delta p\text{CO}_2$ (Figure 5e shown by the red oval and the differences between the methods in Figure A3a,b). A comparison between the SVR and RFR trained with SOCAT v2 and v3 further eliminates the use of different training datasets as primary sources of difference, where methodology
is a higher order driver of difference (Figure A2). The lack of this feature in the eastern Atlantic sector of the Southern Ocean in SVR and RFR estimates suggests that this is a function of the initial clustering step in the SOM-FFN. The clustering process separates $p\text{CO}_2$ observations into clusters that are not restricted in time and space (Landschützer et al., 2014). This allows the SOM-FFN to "transfer knowledge" from a remote location (even outside the Southern Hemisphere) if proxies are similar to the Southern Ocean. This knowledge transfer assumes that the relationship between $p\text{CO}_2$ and the measured proxies is globally
consistent. Moreover, there is the assumption that all $p\text{CO}_2$ variability (within a cluster) can captured by the measured proxies. This assumption is not made when using coordinates or regional subsets as locations are isolated, but there is then the potential loss of knowledge from remote locations. This question will be addressed further in the discussion on the use of coordinate variables as proxies of $\Delta p\text{CO}_2$.

Another difference between $\Delta p\text{CO}_2$ estimates is the tendency for SVR to overestimate $\Delta p\text{CO}_2$ relative to the RFR and
SOM-FFN approaches, particularly in the PFZ and AZ where winter data is sparse (Figure 6)b,c. We attribute this to the SVR's sensitivity to outliers, determined by the fact that the cost function penalises outliers heavily (Equation 5). In context of the SOCAT v3 dataset, the algorithm may treat the sparse winter data as outliers. This is due to the fact that sparse winter measurements of $\Delta p\text{CO}_2$ are positive, while the abundant summer measurements are negative Metzl et al. (2006); Lenton et al. (2013). This may then be a positive realisation of a methodological attribute that is typically considered a weakness.
Conversely, RFR winter estimates of $\Delta p\text{CO}_2$ are often lower than the SOM-FFN and SVR estimates, again in the AZ and PFZ (Figures 5g–i and 6b,c). This may be due to the method's resilience against outliers, which could be due to two attributes (Louppe, 2014). Firstly, outliers are less likely to dominate the feature space with the use of bootstrap aggregation as these points will be sampled less frequently. Secondly, individual decision trees regress values by using the average of samples in a terminal node (or leaf), where the minimum number of samples per terminal node is set by the user. This second attribute

means that estimates will never be outside the bounds of the minimum and maximum of the training dataset, thus leading to conservative estimates (as shown in Figure 2b). The differences between the methods shown in Figure 6 could be a good case for an ensemble approach, where the strengths of one model compensate for the weakness of another. This is assessed in the synthetic data and will be discussed further.

These attributes may also be the reason for the differences in the magnitudes of the autumn peak in $\Delta pCO_2$ in the SAZ and PFZ. Mechanistically this peak could be attributed to a sharp increase in cooling leading into winter, resulting in increased solubility of $CO_2$ and thus a sharp reduction of $\Delta pCO_2$ (Metzl et al., 2006; Takahashi et al., 2002). Deeper mixing of the water column shortly thereafter would entrain $CO_2$ rich waters, thus increasing $\Delta pCO_2$ (Lenton et al., 2013). However, the trend for this peak to shrink in the SAZ and PFZ for all methods suggests that this may an artefact that is specific to the SOCAT dataset.

## 4.2 Synthetic data experiments

In this section we discuss the outcomes of the two experiments performed on the synthetic dataset (BIOPERIANT05 model output). The first experiment addresses the efficacy of including coordinate variables as proxies of $\Delta pCO_2$. This is done by running two implementations RFR and SVR: without coordinates as proxies, and with coordinates as proxies. The second experiment addresses the impact that the SOCAT dataset, biased in both space and time has on $\Delta pCO_2$ estimates.

### 4.2.1 Coordinate variables improve estimates

This topic has to some extent already been mentioned in the discussion of the observational data, where pointed out the case for and against the inclusion of coordinate variables as proxies for $\Delta pCO_2$. If coordinates are not included there is the benefit of potential information transfer from remote parts of the domain, but this assumes that the satellite observable proxies (and assimilative model output) constrain $\Delta pCO_2$ in a globally consistent way. If coordinates are included the information transfer is lost and the assumption is made that the proxy variables are not able to constrain $\Delta pCO_2$ in a globally consistent manner.

The results of this experiment show that coordinates improve estimates of $\Delta pCO_2$ with better RMSE scores for both SVR and RFR (Table 3). We are thus in favour of the second hypothesis that the available proxies cannot sufficiently constrain $\Delta pCO_2$ without coordinates. A two step clustering approach, such as SOM-FFN, may be able to achieve comparable results without coordinates, but this would have to be tested with that specific method. However, this may also lead to trends in the data that may be artefacts of remote knowledge transfer, as potentially seen in the observational data (Figure 5f).

An important outcome of this experiment is that the inclusion of coordinates improves the seasonal phasing of the methods (Figure 8a,b). It is critical for the empirical methods to correctly estimate the phasing of $\Delta pCO_2$ as the seasonal cycle phasing may be a useful indicator of anthropogenic driven changes to the marine carbonate system.

One of the assumptions in these synthetic data experiments is that the models are, to some extent, representative of the variability in the observed ocean. However, the BIOPERIANT05 output does not achieve this, with a standard deviation of 19.80 µatm for synthetic SOCAT v3 data compared to 38.20 µatm of the gridded SOCAT v3 observations (according to Southern Ocean as defined by Fay and McKinley 2014). This could be a cause for concern. However, we believe that this creates an even a stronger case for the use of coordinates as proxy variables. The increased variability in the observations could

be due to processes that deterministic models can not yet constrain due to our lack of understanding of the marine carbonate system (Lenton et al., 2013; Mongwe et al., 2016).

### 4.2.2 SOCAT biases

The lack of winter $pCO_2$ data is a problem throughout the mid and high latitude oceans, but is particularly severe in the Southern Ocean (Bakker et al., 2016), but the impact of the lack of data in the Southern Ocean is not known. Moreover, the efficacy of various methods to fill this large temporal gap is unknown (Rödenbeck et al., 2015). Here we show that there is a considerable impact in this synthetic data environment, but the effect of the sampling bias is perhaps smaller than we would have anticipated. Both methods are able to estimate the spatial distribution and the seasonal cycle of $\Delta pCO_2$ with relative accuracy (Figures 8 and 7 and Table 3). This could be due to two factors.

Firstly, winter data is less variable than summer data and requires less sampling. Mechanistically, this is a likely scenario. In summer $\Delta pCO_2$ is spatio-temporally heterogeneous in the Southern Ocean due to the uptake of $CO_2$ by phytoplankton (Metzl et al., 2006; Bakker et al., 2008; Thomalla et al., 2011; Chierici et al., 2012; Lenton et al., 2013). The drivers of phytoplankton are complex due to the co-limitation of light and iron (as a micronutrient) in the Southern Ocean (Boyd and Ellwood, 2010; Thomalla et al., 2011; Tagliabue et al., 2014). This complexity would require more sampling, perhaps additional proxies or increased spatial resolution to capture the variability of $\Delta pCO_2$. Conversely, processes driving winter $\Delta pCO_2$, namely the interaction of mixing and buoyancy, act on larger scales, potentially leading to less spatio-temporal heterogeneity. However, the lack of observations means that we simply cannot know with certainty. This makes a strong case for autonomous sampling platforms to the Southern Ocean's winter sampling gap. The SOCCOM float project may soon yield such measurements with pH derived estimates of $pCO_2$ (Russell et al., 2014; Johnson et al., 2017; Williams et al., 2017).

Secondly, the model used to generate the synthetic data may not be representative of the Southern Ocean. This has been discussed in the previous section, but here, rather than increasing our confidence, it diminishes our confidence in the result. Studies have shown that process models are not able to accurately represent the seasonal cycle of $CO_2$ in the Southern Ocean (Lenton et al., 2013; Mongwe et al., 2016). Moreover, $\Delta pCO_2$ is often driven by processes that are not representative of observations (Mongwe et al., 2016).

The most likely scenario is likely a combination of these two factors, where winter data is in fact less variable than summer data, but the error is larger than the experiment shows due to incomplete knowledge of the processes that describe $pCO_2$ in the process models.

### 4.2.3 The best method: the ensemble average

The synthetic data also allows us to compare the two methods relative to each other, in the context of the SOCAT v3 data. The data show that the RFR method performs better than the SVR (trained with coordinates as proxies) with respective out-of-sample RMSEs of 5.76 and 6.19 µatm (Table 3). However, it is the average of these two methods (ensemble mean) that achieves the lowest RMSE (5.36 µatm), albeit marginal. The time series in Figure 8c shows that the improvement may come from the period 1998 to 2000, when RFR is plagued by underestimation of the sink strength, while SVR overestimates the sink

strength. This supports the notion that the strengths and weaknesses of these two methods compliment each other. Moreover, it supports the merit of multiple approaches and further development of empirical methods for the estimation of $\Delta p\text{CO}_2$.

## 5 Conclusions

In this study two empirical methods (SVR and RFR) are presented as alternative (and perhaps complementary) $p\text{CO}_2$ gap filling methods. These algorithms are established in other fields, but have not been applied for the estimation of surface ocean $\Delta p\text{CO}_2$. We apply the methods to the Southern Ocean where the paucity of ship based measurements during winter is one of the major challenges. The SOCAT v3 dataset was co-located with assimilative model output and satellite measurable proxy variables to create a training dataset (Bakker et al., 2016). These estimates were compared with the SOM-FFN approach by Landschützer et al. (2014).

We found that the SOM-FFN method outperformed the new approaches with lower RMSE estimates than the RFR and SVR. The RFR performed comparably to the SOM-FFN approach when compared with the SOCAT v2 dataset, with which SOM-FFN was trained. The increase in the number of measurements in the highly variable coastal ocean between SOCAT v2 and v3 leads to increased RMSE values, particularly in the Subantarctic Zone (SAZ). Despite accounting for the increase in coastal data, the SOM-FFN still outperformed the SVR and RFR approaches in the SAZ. We attribute this to the methods ability to cluster the training data into regions of different modes of variability to which individual regressions are then applied. The SVR method performed poorly due to its inability to adapt to various modes of variability, while the RFR is intrinsically much more flexible, thus performed well in fitting the training data.

There was good agreement amongst the three methods with respect to the overall trend of $\Delta p\text{CO}_2$, but there were also differences. The primary difference was in the the Atlantic sector of the SAZ, where the SOM-FFN overestimated $\Delta p\text{CO}_2$ relative to the other methods. This is likely due to remote knowledge transfer within a data sparse cluster; however, we cannot identify this as right or wrong due to the lack of data in this region. Other differences were due to intrinsic attributes of the methods: SVR was sensitive to outliers resulting in relatively large winter $\Delta p\text{CO}_2$ estimates – potentially a desirable feature for sparse winter data; RFR underestimated $\Delta p\text{CO}_2$ relative to the other methods due to its robustness to outliers.

To test the efficacy of these methods, they were applied to a synthetic dataset (process model output). Two major questions were asked: 1) what is the efficacy of including coordinate variables (time, latitude and longitude) as proxy variables? 2) What is the impact of sampling biases in the SOCAT v3 dataset? The results showed that including coordinate variables improved the estimates of $\Delta p\text{CO}_2$ for SVR and RFR. Moreover, the phasing of the seasonal cycle was also improved with the inclusion of coordinates. The second experiment showed that there is only a small bias in the estimates of $\Delta p\text{CO}_2$; however, the inability of process models to represent Southern Ocean $\Delta p\text{CO}_2$ variability accurately places uncertainty on this result.

Lastly we show that while the RFR approach outperforms the SVR approach, the ensemble mean of the two methods scores better than either individual methods. This motivates for continued research on methods that complement each other in strengths and weaknesses.

*Data availability.* The data are available at (https://figshare.com/s/dd034ad593cfd8c5188a)

## Appendix A: Comparison of SOCAT v2 and v3

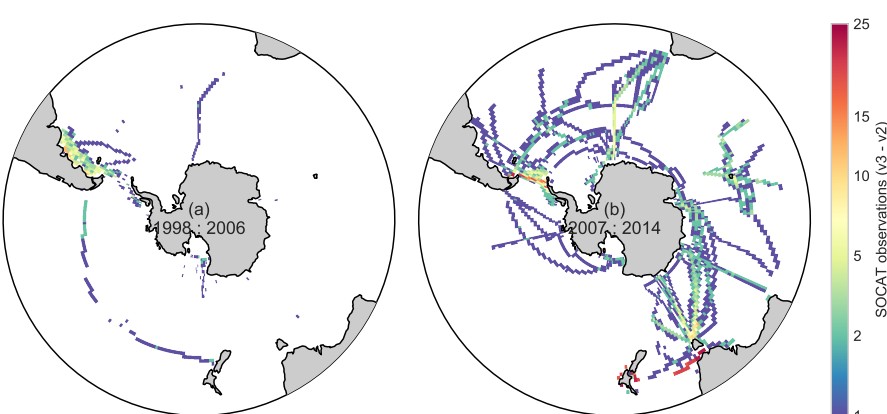

**Figure A1.** The increase in the number of observations between SOCAT v2 and SOCAT v3 for two periods: (a) 1998 through 2006, and 2007 through 2014.

One of the shortcomings of this study is that the SOM-FFN method used SOCAT v2 as a training dataset, while the SVR and RFR methods were trained with SOCAT v3. Figure A3 shows, there is a marked difference between the two datasets. Impor-
5  tantly, the increase in the number of observations between 1998 and 2006 between SOCAT v2 and v3 are almost exclusively in the Argentine Sea.

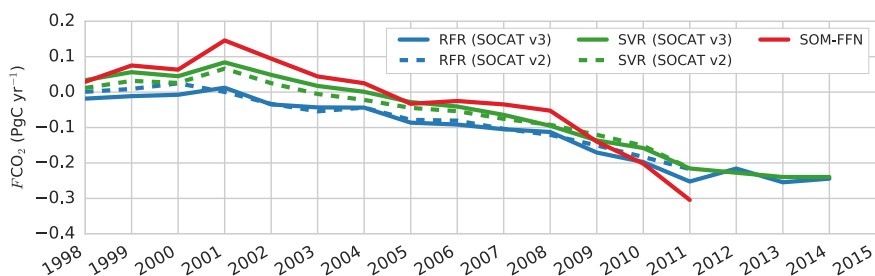

**Figure A2.** Comparison air-sea $CO_2$ flux RFR and SVR when trained with SOCAT v2 and v3. The SOM-FFN method, trained with SO-CAT v2, is also shown. The figure demonstrates that methodology plays a larger role in determining the outcome of the estimate than the availability of data (for these two methods).

These differences may have an impact on the estimates of $\Delta p\text{CO}_2$. To test this, the methods were implemented as explained in Section 2.4 with the exception that RFR and SVR methods were trained with both SOCAT v2 and v3. Figure A2 shows that, on average, there is a larger difference between the RFR and SVR methods than the different training datasets.

The differences between the different methods are shown in Figure A3. Figures (a) and (b) show that the SVR and RFR methods estimate a stronger sink in the Atlantic sector of the SAZ. Here (Figure A3b) the tendency of the SVR method to estimate strong outgassing south of the Polar Front relative to SOM-FFN and RFR is also seen. Conversely, the RFR, on average, underestimates $\Delta p\text{CO}_2$ south of the Polar Front.

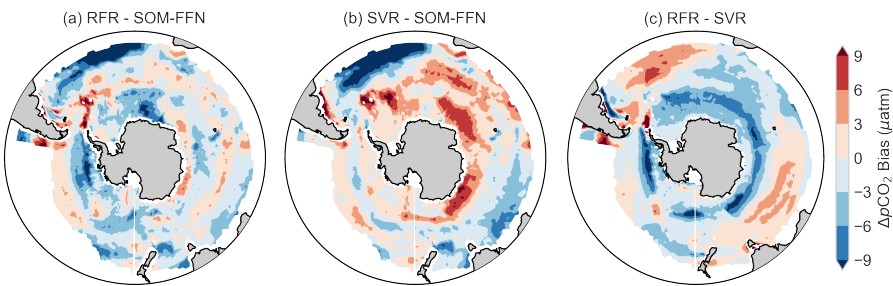

**Figure A3.** The differences between annual averages of each of the approaches for the period 1998 to 2006: (a) RFR – SOM-FFN; (b) SVR – SOM-FFN; (c) RFR – SVR.

## Appendix B: Synthetic data experiments

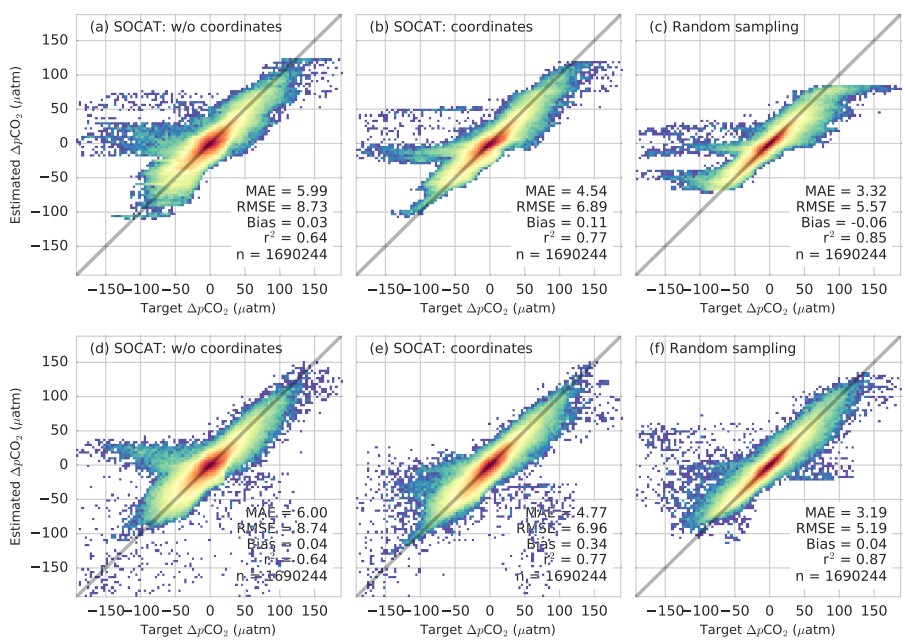

**Figure B1.** Two dimensional histograms for the distributions of out-of-sample estimates of $\Delta pCO_2$ relative to target $\Delta pCO_2$ (BIOPERI-ANT05). The top row (a–c) shows estimates made by RFR and the bottom row (d–f) shows estimates of SVR. The first column (a,c) shows those estimates trained SOCAT v3 locations without coordinate variables (time, latitude and longitude) as proxy variables and the second column (b,e) shows those with coordinate proxies. The last column (c,f) shows estimates trained with random locations (uniform in time and space) with coordinate proxies. The metrics are shown on each plot where MAE and RMSE are Mean Absolute Error and, Root Mean Squared Error respectively. $n$ shows the number of observations in the estimate.

*Acknowledgements.* This work is part of a PhD funded by the ACCESS program. The authors would like to thank Marina Lévy for use of the BIOPERIANT05-GAA95b model data. This work was partly enabled by the Centre for High Performance Computing (CSIR).

The Surface Ocean CO2 Atlas (SOCAT) is an international effort, endorsed by the International Ocean Carbon Coordination Project
5 (IOCCP), the Surface Ocean Lower Atmosphere Study (SOLAS) and the Integrated Marine Biogeochemistry and Ecosystem Research program (IMBER), to deliver a uniformly quality-controlled surface ocean CO2 database. The many researchers and funding agencies responsible for the collection of data and quality control are thanked for their contributions to SOCAT.

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
