# Peer review of "Empirical methods for the estimation of Southern Ocean CO2: Support Vector and Random Forest Regression"

_Biogeosciences, 2017_

## Referee Comment (RC1) · Anonymous Referee #1 · 18 Jun 2017

**Review of Gregor et al: Empirical methods for the estimation of Southern Ocean CO2: Support Vector and Random Forest Regression. Submitted to Biogeosciences**

**Summary:**

Gregor and colleagues present two novel data interpolation methods used to reconstruct the sea surface partial pressure of CO2 and further the air-sea exchange of CO2 from 1998-2014. In a first step, the authors compare their newly derived estimates to an existing estimate (SOM-FFN by Landschützer et al.) for 3 regions as defined by Fay and McKinley, whereas in the second step the authors combine their methods with output from a process model to estimate errors for both the domain where observations do and do not exist. In their work the authors discuss differences between their new estimates and the SOM-FFN method and they further suggest that the uncertainty of current estimates is likely underestimated.

**Strengths:**

The authors provide two complementary new estimates shedding light on the Southern Ocean as a carbon sink, as well as current limitations in estimating this sink. The study critically reflects on past research and discusses differences to their new estimates. While this paper is largely methodological, it also briefly discusses seasonal cycle and trends over the 1998-2014 period. Another strength of the paper is that the authors combine data-based and process–based modelling approaches to test whether the current observational network is sufficient to estimate the Southern Ocean carbon sink within a certain uncertainty level. The topic nicely fits the scope of the journal and I believe this paper is certainly of interest to the wider Biogeosciences readership.

**Weaknesses:**

During my review, I have encountered a few things that need clarification from the authors. They are listed from the most to least concerning. A full list of comments with line-numbers can be found at the end of this document:

- Method section: The weakest bit of this paper certainly is the methods section at the moment. Particularly the 2 approaches are explained to briefly. It is very difficult to follow with many new terms being introduced but not explained, e.g:
  " A few slack variables ( ) are allowed, within the limits of a slack parameter" – what are "slack variables" and "slack parameter"?
  " versatile by mapping X onto a higher dimensional feature space using an interchangeable kernel" – feature space? interchangeable kernel?
  " decision trees" – to the average BG reader a tree has leaves or needles …
  " bagging" – the meaning is not clear
  "K-fold cross-validation" – again, please explain what this means
  Without knowing all these terms the reader is lost and understanding a method means trusting a method.

- Validation, comparison: It is disappointing that the authors only provide the RMSE MAE and r2 in the manuscript for the entire period, i.e only one number. Many statements in the text do require a more thorough analysis. E.G section 4.1: " One of the most marked differences is the weaker sink estimated by the SOM-

FFN method in the SAZ (Figure 4)." – Figure 4 shows that the difference between the estimates is e.g. larger in the earlier analysis years – a error/RMSE/r2 analysis per year would be interesting and make a stronger case. Furthermore, it would be very interesting how the error/RMSE/r2 varies with data density, both in time and in space.

- The usage of space and time coordinates: Firstly, I am not surprised that additional data result in a smaller error, as they add additional degrees of freedom. Secondly, after reading the methods section a was puzzled why they were included? In the end, on page 12 line 21 I found the statement: " This implies that the available proxy variables are not able to capture the variability of pCO2 ." pCO2 is not affected by time and space, but by the environmental conditions reflected in proxies such as SST or biology. Space and time are in this case only placeholders for unknown proxies. This needs to be better discussed up-front.

**Recommendation:**

I have found this study to be very interesting and to be of value to the readership. The results are interesting, as the Southern Ocean carbon uptake is a hot topic at the moments and the question arises to what extend we can estimate the sink using the sparse observations we have. The study shows to what extend methodological differences add to the uncertainty in estimates and it tries to show that ship-based estimates underestimate the overall uncertainty. While I do believe the study should be published I don't think it is publication-ready at the moment. The 3 points raised above are of major concern and at least points 1 and 2 need to be added and point 3 at least discussed before publication. I therefor **recommend major revisions of the manuscript**.

**Specific and minor comments to the text:**

Page 2 line 9: "were" – I suppose "where"

Page 2 line 10: "interannual pCO2 trends" – interannual trends? I suppose you mean interannual variability, otherwise please clarify

Page 2 lines 16-18: This statement is right but wrong: Rödenbeck et al indeed did argue that there is a lack of independent ship-based observations in the SO which prohibit an independent comparison – hence right. However, e.g. Landschützer et al 2015 used for their trend analysis also an atmospheric inverse estimate which is based on independent, namely atmospheric, observations – hence wrong. So, in combination with the text above this statement is misleading.

Page 4 line 8: "gridded observations" – I don't think – not even for the sake of brevity – you can call data from an assimilation model (ECCO2) "gridded observations"

Page 5 line 5-6: You claim that log10 normalisation of CHL and MLD leads to normal distribution, but I doubt that – I suspect it rather comprised a fairly normal distribution in the center with long tails.

Page 5 lines 9-10 and following: see major comment above. A bit more discussion is needed what these coordinates represent in terms of CO2 predictors.
Page 5 line 25 and following: The methods are hard to follow. Too many unknown and specific wording is used (see major comment above).

Page 7 lines 14-15: why Nightingale? there are newer transfer velocity estimates from Wanninkhof et al. (2013, 2014) using CCMP?

Page 8 Figure 3: It is confusing that the SOM-FFN method is called "SOM" here – please don't change abbreviations throughout the manuscript.

Page 9 Figure 4: In all the following text the difference between the lines is discussed, but not that they are based on different datasets, i.e. SOCATv2 and SOCATv3. It is certainly plausible that the availability of data in SOCAT also affects the difference? I suggest to discuss this also in the main text.

Page 11 lines 5-12: This is very vague. Firstly, the authors have not properly calculated uncertainties for each region and timestep. Secondly – as mentioned above, the discussion is missing the difference between SOCATv2 and SOCATv3. How many new data are included in SOCATv3 and where? Could this add to the difference? Thirdly, the statement about the influence of the tropics is vague.

Page 12 lines 15-16: I suppose discontinuity at a cluster, or biome border is a sign of bad model quality as well. In 2 adjacent biomes, that are well sampled, I would expect no hard border, whereas in more poorly reconstructed biomes this border effect is more prominent. However, continuity is no sign of quality, but rather comprises a "prettier picture".

---

## Referee Comment (RC2) · Prof GRUBER (Referee) · 29 Jul 2017

**1 Summary**

Gregor et al. introduce and test two new statistical methods to interpolate the relatively sparse surface ocean pCO$_2$ data in the Southern Ocean to produce time and space continuous distributions of this quantity, from which the air-sea CO$_2$ flux can be computed. Underlying both methods, i.e., Support Vector Regression (SVR) and Random Forest Regression (RFR), are machine-learning approaches, wherein non-linear relationships are established between a set of observed independent variables and the

target property surface ocean pCO$_2$. In addition to using standard evaluation metrics to determine the quality of the fits, the authors also use output from a model where the "truth" is known, permitting them also to assess the contribution of the sparse sampling. Gregor et al. find that both models perform overall well and comparable with some of the best performing models published. The RFR model tends to have overall a lower root-mean-square-error (RMSE) than SVR, although the SVR was performing better in a test where the predictability beyond the training data set was assessed. The ensemble of the two methods confirms recent findings that the Southern Ocean carbon sink appears to have recovered from a period of low uptake during the late 1990s by having strengthened substantially all the way into the 2010s.

**2 Evaluation**

This is a very valuable and important contribution to the long-standing quest of quantifying the ocean uptake of CO$_2$, particularly in a region that is undeniably key for determining the global uptake. While there exist already nearly two handful of approaches in the literature that pursue the same objective, the sophistication of the approach taken by Gregor et al., the depth of their analyses, and the independent checking of the result via synthetic data makes this a novel and highly relevant study. Therefore, I am very supportive of this manuscript and would love to see published as soon as possible. But before giving the final green light, there are a couple of major comments/concerns that I would like the authors to carefully consider.

- Deepen analysis: While the manuscript is relatively thorough in the description of the two methods (with exceptions - see minor comments below), I find that the evaluation part has quite some room for an extension and some deepening. In particular, I am missing a thorough analysis of the residuals in time and space.

- Extend scientific discussion of method: The paper would benefit substantially from an extension of the scientific discussion of the pros and cons of the method. Many issues are currently mentioned and investigated, but few of them are really discussed to the necessary level of detail and finality. Examples include the inclusion of the spatial variables in the regression, which is tested, but then only partially further investigated. Another good example is the more limited predictability of the RFR relative to the SVR. Finally, with this new method needing to compete with a range of already existing methods, the authors needs to demonstrate more clearly why it is better. I understand that these are difficult issues to discuss, and that it is likely not possible to give a definite answer. But it would behoove the authors well to push the manuscript as far as possible in this direction.

- Deepen scientific analyses and discussion of results: As it stands, the paper focuses nearly entirely on the method, and leaves only very little room for the scientific findings. This is a shame, in my opinion. I think that there is enough room in the manuscript to add a few more scientific analyses to the paper and to discuss them thoroughly. One example is the seasonal cycle, which differs quite substantially between the different estimates and is hugely important for determining the annual $CO_2$ sink.

- Language/Grammar: There are several places where the writing can be improved and be made more concise and precise. Further, the manuscript contains a number of grammatical/typographic errors that should be eliminated before the resubmission.

**3  Recommendation**

I very much support the publication of this manuscript, but it requires a moderate revision before I am ready to fully endorse it. The revision needs to focus on the three

major issues identified above, i.e., more thorough analysis and discussion of method, and extended discussion of results.

**4  Minor comments**

Abstract, p1, line 5: I suggest to add the source of the data already here, i.e., to write "The methods are used to estimate DpCO2 in the Southern Ocean based on SOCAT V3... ".

Abstract, p1, line 6: Typo. Change "The RFR as able" to "The RFR is able"

Abstract, p1, lines 6-7 and elsewhere: I don't think that there is a statistically significant difference in the RMSE between 12.26 and 12.97 $\mu$atm. Please rephrase.

Abstract, p1, line 8: "modeled environment". The commonly used expression here is "synthetic data".

Abstract, p1, line 9: "achieved". Not sure that this is the best expression, since one commonly tries to achieve something that is desirable. I am not sure that having a higher error is a desired outcome. Perhaps simply write "have".

Abstract, p1, line 11: Add "a" to ratio, i.e., to read "with a lower ratio".

Abstract, p1, general: Following up on my major comments (ii) and (iii), I think that also the abstract could benefit from a reshuffling with a bit more text devoted to the discussion of the methods and how they compare to others, and a bit more text about the results.

Introduction, p1-3: general: The introduction reads well and contains the most important pieces, but I would love to see a bit more material with regard to the particular strengths and weaknesses of the existing methods. As it stands, it does not become clear to the average reader why we need yet another set of methods to interpolate

the sparse data. This also helps to set up the later discussion on how this new set of methods stacks up against the existing ones.

Data and methods: p4, line 1-4: It would be much cleaner if you used the same biomes for the synthetic data as for the real data. Of course are the model boundaries somewhat different if one used the same criteria as used by Fay and McKinley, but this really should not matter much. Much more relevant is that you use the same approach when using the synthetic and the real data, so that you can really draw conclusions from one approach to the other. I strongly suggest to reconsider this choice.

Model data: p5, line 2: "resampled to ... monthly averaged resolution" This likely adds quite some smoothing to the data, something that does not really exist in the observations. Although the latter have been binned to 1x1 dg and month of the year, but many grid cells contain only a few observations, and therefore do not really represent a monthly average. Why not spot sampling the model following the sampling scheme of the observational programs?

Data transformation: p5, line 5 (and elsewhere). "There are several transformations that are applied..." This is awkward and can be simplified (and improved) to "The input data are transformed..."

Data transformation: p5, lines 9-10: "This then raises the question..." I found this somewhat confusing. I suggest that you simply describe what you did in the method section, i.e., that your standard model includes the spatial coordinates, but that you also tested a version without them, and then have a more thorough discussion in the discussion section.

Data transformation: p5, lines 16-20: I suggest to add here somewhere the time period that these data cover.

Empirical methods: p5, line 21, Data are plural. Thus "The data are split..."

SVR: p5, line 26: "The formulation of the SVR is such..." Awkward writing. I suggest to

simplify this to "The cost function of the SVR minimizes ..."

SVR: p6, lines 1-7: I suggest to add a bit more text here to better explain the SVR, and in particular, to better explain the particular parameter choices.

RFR: p7, lines 1-6: As above, I also suggest here to better explain the method and the parameter choices.

RFR: p7, line 9 "The out-of-bag error is used to select the hyper-parameters..." This is extraordinary cryptic. Please explain better.

CO2 fluxes: p7, lines 14-16. "calculated". This expression is used three times in a row in a very repetitive manner. This makes it boring and hard to read. Please reformulate.

Results, p8, lines 3-6: This connects to my first major comment. In my opinion, this section needs to be substantially extended and strengthened. A comparison of correlation coefficients and RMSE is insufficient in my opinion. I would like to see an analysis of the pattern and structure of the residuals in time and space. I also would like to see the biases and perhaps a few other metrics.

Results, p8, line 5: "slightly better...". I don't think that this statement holds up to further scrutiny. With a measurement error of about 1 $\mu$atm and data that are distributed in time and space anything but random, I don't think that this difference is significant. To me, all one can say is that the two results are comparable in performance.

Results, p8, Figure 3: I would love to see also the annual mean figure and its discussion added to the results section.

Results, p9, line 9: "Estimates are higher..." but also elsewhere This is a result that is picked up here, but it is not really discussed later on. This is just one example of a few such mismatches between results and the later discussion section.

Results, p10, lines 15-16 "Out-of-bag error" and "Out-of-sample error". These terms are uncommon and thus need to be carefully defined and later repeated in order for

the average reader to be able to follow the arguments.

Results, p11, "These results suggest that estimates would benefit from the inclusion of coordinates". This statement is problematic for various reasons. First, such a conclusion should not really be part of the results section. Second, I don't really buy the argument, since almost by definition, the inclusion of additional independent variables tends to improve the fit, i.e., it increases the degrees of freedom of the problem at hand. This does not imply an increase in predictability or a true increase in "knowledge", as tested, for example through an analysis of the Akaike information criterion (AIC). Personally, I also oppose the inclusion of such variables, as they do not include any process information, and, in fact, suppress the establishment of knowledge transfer between regions of similar dynamics, but distant in time/space. I suggest to reconsider this choice and conclusion.

Discussion, p11, line 6: "weaker sink". This is not really obvious from Figure 4. I suggest to add a figure showing the annual mean DpCO2 including the differences between the different estimates. With such a figure, the whole paragraph becomes much easier to follow.

Discussion, p11, line 19: "sparse winter data". This is certainly a valid hypothesis, but couldn't the authors use the synthetic data to test this hypothesis?

Discussion, p12, line 1: "Ensemble estimate". This is not an unreasonable assumption, but it is again one that could be easily tested with the synthetic data.

Discussion, p12, line 15: "additional complexity of dealing with DpCO2 discontinuities" It turns out that this is a very small issue. You can test this by comparing the smoothed with the raw version in the pCO2 data sets provided by Landschützer et al. See http://cdiac.ornl.gov/oceans/SPCO2_1982_2011_ETH_SOM_FFN.html.

Discussion, p12, lines 17-30: The conclusion stated on page 11 about the inclusion of a spatial variable should come, at the earliest here.

Discussion, p12, in general: There are many other things that need to be discussed here (see also my second major comment above).

Discussion, p13, line 2, "Tuning the algorithm..." This sentence needs to be embedded better in order for it to make sense to the average reader.

Discussion, p13, section 4.4. "Trends of ensemble estimates". This section and related ones needs to be substantially strengthened. As it stands, this small section is not much more than a teaser. This should not be.

Conclusion, p13, line 32, "from satellite proxies..." This is not quite correct, since SSS, MLD, and atm. $CO_2$ stem from other sources. Please reformulate.

Conclusions, p14, lines 4-10: Some of these conclusions are not really that evident from the results provided earlier. This has a lot to do with the results section not having made the point well enough.

Data availability, p14: I think it would be much better if the data were hosted by an international database such as CDIAC (in the future NCEI) or Pangaea.

Nicolas Gruber July 2017

---

## Author Response (AR1)

**Reviewers initial comments are in dark blue Responses to the comments are in green**

**Response to both reviewers**

We would like to thank the reviewers for the comprehensive and constructive feedback on the manuscript. We feel that the comments that they made have contributed to a much better manuscript.

Some of the recommended changes were fairly large and thus the manuscript, primarily the results and discussion, have changed significantly. As recommended by both reviewers, we have deepened both the analyses and the discussion. In summary:

- A deeper analyses of the observational estimates assesses the performance based on the RMSE scores (and other metrics) and then assesses the difference between the estimates.
- The synthetic data experiments have been formalised. There are now two primary experiments: 1) what is the impact of including coordinates as proxies on the estimates; 2) what is the effect of the sampling biases in SOCAT v3. We also find that the ensemble mean of  $\Delta$ pCO2 scores better than the individual methods.
- The discussion is structured after the results (two points above), but now goes much more in depth.
- The final figure focussing on the trends of the fluxes has been removed. This is due to the fact that the manuscript is now much longer and the trend analysis would distract the reader from the primary goal of the study; which is to introduce methods and the synthetic data experiments.
- These results will be published in a future publication.

The remaining comments have been addressed to each specific reviewer.

We hope that these changes make the manuscript suitable for publication.

**Response to R1**

Weaknesses

Methods: The weakest bit of this paper certainly is the methods section at the moment. Particularly the 2 approaches are explained to briefly. It is very difficult to follow with many new terms being introduced but not explained, e.g: "A few slack variables () are allowed, within the limits of a slack parameter" – what are "slack variables" and "slack parameter"? "versatile by mapping X onto a higher dimensional feature space using an interchangeable kernel" – feature space? interchangeable kernel? "decision trees" – to the average BG reader a tree has leaves or needles ... "bagging" – the meaning is not clear "K-fold cross-validation" – again, please explain what this means Without knowing all these terms the reader is lost and understanding a method means trusting a method.

We have addressed this weakness by including more detail about each of the methods. This includes the basic formulation for SVR and RFR. The terms are also now explained more explicitly.

Validation, comparison: It is disappointing that the authors only provide the RMSE MAE and r2 in the manuscript for the entire period, i.e only one number. Many statements in the text do require a more thorough analysis. E.G section 4.1: " One of the most marked differences is the weaker sink estimated by the SOM-FFN method in the SAZ (Figure 4)." – Figure 4 shows that the difference between the estimates is e.g. larger in the earlier analysis years – a error/RMSE/r2 analysis per year would be interesting and make a stronger case. Furthermore, it would be very interesting how the error/RMSE/r2 varies with data density, both in time and in space. We have included three regional time series of RMSE for each of the biomes. These include the data density. Note that the RMSE values have also increased as these were previously reported for only the SAZ and FPZ combined. There is also now an analysis of the RMSE in the SAZ included as additional material. This shows that the increase in error in the SAZ is primarily due to increased number of coastal

measurements.

The usage of space and time coordinates: Firstly, I am not surprised that additional data result in a smaller error, as they add additional degrees of freedom. Secondly, after reading the methods section I was puzzled why they were included? In the end, on page 12 line 21 I found the statement: "This implies that the available proxy variables are not able to capture the variability of pCO2." pCO2 is not affected by time and space, but by the environmental conditions reflected in proxies such as SST or biology. Space and time are in this case only placeholders for unknown proxies. This needs to be better discussed up-front.

**This is addressed a little better in section 2.3: Data transformation and derived variables. The paragraph now reads:**

To gain a better understanding of these methods' strengths and weaknesses we implement SVR and RFR in a synthetic data environment. A similar approach was taken by {Friedrich2009} in the North Atlantic, which experienced a similar data paucity to the Southern Ocean in the early 2000's. This idealised environment was also used to estimate the effect of including/excluding certain proxy variables as well as the optimal coverage of cruise tracks to constrain the North Atlantic  $\Delta pCO_2$  adequately. Similarly, we assess the efficacy of including coordinate variables as proxies of  $\Delta pCO_2$  in the empirical methods. In the intercomparison study by proxies typically include, but are not limited to sea surface temperature (SST), chlorophyll-a (Chl-a), mixed layer depth (MLD) and sea surface salinity (SSS); however several methods in the study also include latitude and longitude. While coordinates do not mechanistically impact  $\Delta pCO_2$ , they do help to constrain estimates where the available remote sensing proxies cannot adequately do so. The synthetic data is also used to test the ability of the SVR and RFR to approximate  $\Delta pCO_2$  in the seasonally sparse Southern Ocean.

Specific and minor comments

- Page 2 line 9: "were" I suppose "where" changed to where
- Page 2 line 10: "interannual pCO2 trends" interannual trends? I suppose you mean interannual variability, otherwise please clarify changed to interannual variability
- Page 2 lines 16-18: This statement is right but wrong: Rödenbeck et al indeed did argue that there is a lack of independent ship-based observations in the SO which prohibit an independent comparison – hence right. However, e.g. Landschützer et al 2015 used for their trend analysis also an atmospheric inverse estimate which is based on independent, namely atmospheric, observations – hence wrong. So, in combination with the text above this statement is misleading. *Changed the paragraph completely. The intro should now read better.*
- Page 4 line 8: "gridded observations" I don't think not even for the sake of brevity – you can call data from an assimilation model (ECCO2) "gridded observations" *Corrected*
- Page 5 line 5-6: You claim that log10 normalisation of CHL and MLD leads to normal distribution, but I doubt that I suspect it rather comprised a fairly normal distribution in the center with long tails.
  - Have changed this to "a distribution that closer resembles a normal distribution"
- Page 5 lines 9-10 and following: see major comment above. A bit more discussion is needed what these coordinates represent in terms of CO2 predictors. Page 5 line 25 and following: The methods are hard to follow. Too many unknown and specific wording is used (see major comment above).
   Details are now "fleshed out" on pages 6 and 7
- Page 7 lines 14-15: why Nightingale? there are newer transfer velocity estimates from Wanninkhof et al. (2013, 2014) using CCMP? *Now using Wanninkhof et al., (2014)*
- Page 8 Figure 3: It is confusing that the SOM-FFN method is called "SOM" here please don't change abbreviations throughout the manuscript.
   SOM-FFN is continued throughout manuscript
- Page 9 Figure 4: In all the following text the difference between the lines is discussed, but not that they are based on different datasets, i.e. SOCATv2 and SOCATv3. It is certainly plausible that the availability of data in SOCAT also affects the difference? I suggest to discuss this also in the main text.

We address this issue in two ways: estimates are compared with SOCAT v2 and v3; models trained with SOCAT v2 and v3 are compared – this is presented only in the appendix.

• Page 11 lines 5-12: This is very vague. Firstly, the authors have not properly calculated uncertainties for each region and timestep. Secondly – as mentioned above, the discussion is missing the difference between SOCATv2 and SOCATv3. How

many new data are included in SOCATv3 and where? Could this add to the difference? Thirdly, the statement about the influence of the tropics is vague. Table 2 has been changed to a figure showing the spatial and temporal variability of RMSE for each of the methods. Moreover, there is also a table detailing the average regional RMSE, MAE, bias, r2 and n.

There are also two new figures in the additional materials that address the issue of SOCAT v2 vs v3. We show that the relative majority of points gained in the SAZ in SOCAT v3 are in the Argentine sea – a region of high complexity. The tropics point has been changed to a discussion around "remote knowledge transfer" and this should now be much clearer.

• Page 12 lines 15-16: I suppose discontinuity at a cluster, or biome border is a sign of bad model quality as well. In 2 adjacent biomes, that are well sampled, I would expect no hard border, whereas in more poorly reconstructed biomes this border effect is more prominent. However, continuity is no sign of quality, but rather comprises a "prettier picture".

*Removed the statement about the discontinuity of clusters as Reviewer 2 pointed that this is a trivial issue to solve.*

**Response to R2**

Evaluation

- Deepen analysis: While the manuscript is relatively thorough in the description of the two methods (with exceptions see minor comments below), I find that the evaluation part has quite some room for an extension and some deepening. In particular, I am missing a thorough analysis of the residuals in time and space. The analyses have been extended significantly. The analysis around the RMSE estimates have been extended and the differences between the methods are now investigated in full.
- Extend scientific discussion of method: The paper would benefit substantially from an extension of the scientific discussion of the pros and cons of the method. Many issues are currently mentioned and investigated, but few of them are really discussed to the necessary level of detail and finality. Examples include the inclusion of the spatial variables in the regression, which is tested, but then only partially further investigated. Another good example is the more limited predictability of the RFR relative to the SVR. Finally, with this new method needing to compete with a range of already existing methods, the authors needs to demonstrate more clearly why it is better. I understand that these are difficult issues to discuss, and that it is likely not possible to give a definite answer. But it would behove the authors well to push the manuscript as far as possible in this direction.

The results and discussion have been extended significantly. The synthetic data experiments have been formalised and are now discussed fully.

• Deepen scientific analyses and discussion of results: As it stands, the paper focuses nearly entirely on the method, and leaves only very little room for the scientific findings. This is a shame, in my opinion. I think that there is enough room in the manuscript to add a few more scientific analyses to the paper and to discuss them thoroughly. One example is the seasonal cycle, which differs quite substantially between the different estimates and is hugely important for determining the annual CO2 sink.

As stated above, the scientific analyses have been deepened and we feel that the manuscript is now more complete.

• Language/Grammar: There are several places where the writing can be improved and be made more concise and precise. Further, the manuscript contains a number of grammatical/typographic errors that should be eliminated before the resubmission. *Changed as recommended in the specific comments below*

**Specific and minor comments**

 Abstract, p1, line 5: I suggest to add the source of the data already here, i.e., to write "The methods are used to estimate DpCO2 in the Southern Ocean based on SOCAT V3...".

SOCAT added to the abstract

- Abstract, p1, line 6: Typo. Change "The RFR as able" to "The RFR is able" corrected as to is
- Abstract, p1, lines 6-7 and elsewhere: I don't think that there is a statistically significant difference in the RMSE between 12.26 and 12.97 µatm. Please rephrase.

phrase removed – also note that these estimates have changed. The previous estimates were for the SAZ and PFZ biomes only.

- Abstract, p1, line 8: "modelled environment". The commonly used expression here is "synthetic data".
  - synthetic data now used throughout the manuscript
- Abstract, p1, line 9: "achieved". Not sure that this is the best expression, since one commonly tries to achieve something that is desirable. I am not sure that having a higher error is a desired outcome. Perhaps simply write "have". *this has been changed throughout the manuscript*
- Abstract, p1, line 11: Add "a" to ratio, i.e., to read "with a lower ratio". added "a"
- Abstract, p1, general: Following up on my major comments (ii) and (iii), I think that also the abstract could benefit from a reshuffling with a bit more text devoted to the discussion of the methods and how they compare to others, and a bit more text about the results.

A large portion of the abstract has been rewritten to accommodate the reviewer's suggestions

• Introduction, p1-3: general: The introduction reads well and contains the most important pieces, but I would love to see a bit more material with regard to the particular strengths and weaknesses of the existing methods. As it stands, it does not become clear to the average reader why we need yet another set of methods to interpolate the sparse data. This also helps to set up the later discussion on how this new set of methods stacks up against the existing ones.

The introduction has been reformatted to include a motivation for each of the methods as well as the description of the different methods and why these were chosen.

Data and methods: p4, line 1-4: It would be much cleaner if you used the same biomes for the synthetic data as for the real data. Of course are the model boundaries some- what different if one used the same criteria as used by Fay and McKinley, but this really should not matter much. Much more relevant is that you use the same approach when using the synthetic and the real data, so that you can really draw conclusions from one approach to the other. I strongly suggest to reconsider this choice.

The northern boundary of the synthetic data has been changed from 30°S to the boundaries defined by Fay and McKinley (2014).

 Model data: p5, line 2: "resampled to ... monthly averaged resolution" This likely adds quite some smoothing to the data, something that does not really exist in the observations. Although the latter have been binned to 1x1 dg and month of the year, but many grid cells contain only a few observations, and therefore do not really represent a monthly average. Why not spot sampling the model following the sampling scheme of the observational programs?

Change has been implemented to the data, and the text now reads: The synthetic observations are sampled at the model resolution (5-day  $\times 0.5^{\circ}$ ) to resemble the SOCAT dataset. Hereafter all data is resampled to 1.0° spatial resolution and monthly temporal resolution data to match observations.

• Data transformation: p5, line 5 (and elsewhere). "There are several transformations that are applied..." This is awkward and can be simplified (and improved) to "The

input data are transformed..." Changed as recommended

• Data transformation: p5, lines 9-10: "This then raises the question..." I found this some- what confusing. I suggest that you simply describe what you did in the method section, i.e., that your standard model includes the spatial coordinates, but that you also tested a version without them, and then have a more thorough discussion in the discussion section.

This has been introduced briefly in the methods – only the methodology is presented

- Data transformation: p5, lines 16-20: I suggest to add here somewhere the time period that these data cover.
  - This was added at the end of the first paragraph in section 2.1 Gridded Data
- Empirical methods: p5, line 21, Data are plural. Thus "The data are split..." SVR: changed as recommended
- p5, line 26: "The formulation of the SVR is such..." Awkward writing. I suggest to simplify this to "The cost function of the SVR minimizes ..." This section has changed – more detail for each method added at the request of Reviewer 1
- SVR: p6, lines 1-7: I suggest to add a bit more text here to better explain the SVR, and in particular, to better explain the particular parameter choices. *More detail has been added. The cost function has been included.*
- RFR: p7, lines 1-6: As above, I also suggest here to better explain the method and the parameter choices.
   More detail has been added about the RFR, specifically, the theoretical model for a decision tree.
- RFR: p7, line 9 "The out-of-bag error is used to select the hyper-parameters..." This is extraordinary cryptic. Please explain better.
  - This should be clearer with the additional information provided.
- CO2 fluxes: p7, lines 14-16. "calculated". This expression is used three times in a row in a very repetitive manner. This makes it boring and hard to read. Please reformulate.

Restructured as suggested

Results, p8, lines 3-6: This connects to my first major comment. In my opinion, this section needs to be substantially extended and strengthened. A comparison of correlation coefficients and RMSE is insufficient in my opinion. I would like to see an analysis of the pattern and structure of the residuals in time and space. I also would like to see the biases and perhaps a few other metrics.

The results and discussion have been updated with a much more in depth look at the RMSE values for the observational estimates

 Results, p8, line 5: "slightly better...". I don't think that this statement holds up to further scrutiny. With a measurement error of about 1 µatm and data that are distributed in time and space anything but random, I don't think that this difference is significant. To me, all one can say is that the two results are comparable in performance.

This has been changed

Results, p8, Figure 3: I would love to see also the annual mean figure and its discussion added to the results section.
 The image has been changed and now includes the mean state.

- Results, p9, line 9: "Estimates are higher..." but also elsewhere This is a result that is
  picked up here, but it is not really discussed later on. This is just one example of a few
  such mismatches between results and the later discussion section.
  These issues have hopefully been ironed out. The results and discussion have been
  rewritten to a large extent.
- Results, p10, lines 15-16 "Out-of-bag error" and "Out-of-sample error". These terms are uncommon and thus need to be carefully defined and later repeated in order for the average reader to be able to follow the arguments.
   We define the in and out of sample errors adequately and are now used frequently enough for the reader to keep track. The out of bag errors are only referred to briefly
- Results, p11, "These results suggest that estimates would benefit from the inclusion
  of coordinates". This statement is problematic for various reasons. First, such a
  conclusion should not really be part of the results section. Second, I don't really buy
  the argument, since almost by definition, the inclusion of additional independent
  variables tends to improve the fit, i.e., it increases the degrees of freedom of the
  problem at hand. This does not imply an increase in predictability or a true increase
  in "knowledge", as tested, for example through an analysis of the Akaike information
  criterion (AIC). Personally, I also oppose the inclusion of such variables, as they do not
  include any process information, and, in fact, suppress the establishment of
  knowledge transfer between regions of similar dynamics, but distant in time/space. I
  suggest to reconsider this choice and conclusion.

The reviewer makes a valid point. However, the whole point of the synthetic data experiment is to test this. We feel that the new synthetic data experiments should better show the pros and cons of coordinates as proxies. We still find that, in the case of RFR and SVR as implemented in this study, should be included as the current available proxies are likely not fully capturing the variability of  $\Delta pCO2$ .

• Discussion, p11, line 6: "weaker sink". This is not really obvious from Figure 4. I suggest to add a figure showing the annual mean DpCO2 including the differences between the different estimates. With such a figure, the whole paragraph becomes much easier to follow.

This region has now been highlighted with a red oval. This is primarily to avoid too many figures in the manuscript. The differences of summer  $\Delta pCO2$  have been added to the additional materials

- Discussion, p11, line 19: "sparse winter data". This is certainly a valid hypothesis, but couldn't the authors use the synthetic data to test this hypothesis? The manuscript now follows a format of two primary synthetic data experiments, where the first asks what the impact of coordinates as proxies is and the second addresses the issue of sampling bias in the SOCAT dataset
- Discussion, p12, line 1: "Ensemble estimate". This is not an unreasonable assumption, but it is again one that could be easily tested with the synthetic data. We now show, with the synthetic data that the ensemble estimate of RFR and SVR is in fact a better fit to the out-of-sample estimate than the standalone methods.

- Discussion, p12, line 15: "additional complexity of dealing with DpCO2 discontinuities" It turns out that this is a very small issue. You can test this by comparing the smoothed with the raw version in the pCO2 data sets provided by Landschützer et al. See http://cdiac.ornl.gov/oceans/SPCO2\_1982\_2011\_ETH\_SOM\_FFN.html. This has been removed from the discussion
- Discussion, p12, lines 17-30: The conclusion stated on page 11 about the inclusion of a spatial variable should come, at the earliest here. *This topic has been moved to the discussion*
- Discussion, p12, in general: There are many other things that need to be discussed here (see also my second major comment above). *The discussion should now be more comprehensive*
- Discussion, p13, line 2, "Tuning the algorithm..." This sentence needs to be embedded better in order for it to make sense to the average reader. *The discussion has changed this sentence no longer exists.*
- Discussion, p13, section 4.4. "Trends of ensemble estimates". This section and related ones needs to be substantially strengthened. As it stands, this small section is not much more than a teaser. This should not be.
   We removed the section on the trends as it may in fact distract the reader from the already dense material. This will be published in the near future.
- Conclusion, p13, line 32, "from satellite proxies..." This is not quite correct, since SSS, MLD, and atm. CO2 stem from other sources. Please reformulate. Sentence now reads: The SOCAT v3 dataset was co-located with assimilative model output and satellite measurable proxy variables to create a training dataset.
- Conclusions, p14, lines 4-10: Some of these conclusions are not really that evident from the results provided earlier. This has a lot to do with the results section not having made the point well enough.

This has been changed substantially and should no longer contain any surprise results.

• Data availability, p14: I think it would be much better if the data were hosted by an international database such as CDIAC (in the future NCEI) or Pangaea. *This will be hosted by FigShare which has DOI*

**Empirical methods for the estimation of Southern Ocean CO2: Support Vector and Random Forest Regression**

Luke Gregor1,2, Schalk Kok3, and Pedro M. S. Monteiro1

1Southern Ocean Carbon-Climate Observatory (SOCCO), CSIR, Cape Town, South Africa 2University of Cape Town, Department of Oceanography, Cape Town, South Africa 3University of Pretoria, Department of Mechanical and Aeronautical Engineering, Pretoria, South Africa

Correspondence to: Luke Gregor (luke.gregor@uct.ac.za)

Abstract. The Southern Ocean accounts for 40% of oceanic  $CO_2$  uptake, but the estimates are bound by large uncertainties due to a paucity in observations. Gap filling empirical methods have been used to good effect to approximate  $pCO_2$  from satellite observable variables in other parts of the ocean, but many of these methods are not in agreement in the Southern Ocean. In this study we propose two additional methods that perform well in the Southern Ocean: Support Vector Regression (SVR) and

- 5 Random Forest Regression (RFR). The methods are used to estimate  $\Delta pCO_2$  in the Southern Ocean based on SOCAT v3, achieving similar results trends to the SOM-FFN method by Landschützer et al. (2014). The RFR as able to achieve better RMSE (12.26 µatm) compared the SVR (16.04 µatm) and Results show that the SOM-FFN (12.97 approach outperforms the RFR and SVR methods with respective RMSE scores of 14.84, 16.45 and 24.40 µatm). To assess the efficacy of the methods and the limits of the training dataset (SOCAT. However, this is, in part, due to an increase in coastal observations from SOCAT.
- 10 v2 to v3), SVR and RFR are applied in a modelled environment. Again. The success of the SOM-FFN and RFR both depend on the ability to adapt to different modes of variability. The SOM-FFN achieves this by having independent regression models for each cluster, while this flexibility is intrinsic to the RFR methodoutperformed the SVR by a substantial margin. However, both methods achieved higher out-of-sample than in-sample errors, indicating that the. Analyses of the estimates shows that the SVR and RFR's respective sensitivity and robustness to outliers define the outcome significantly. Further analyses on the
- 15 methods were performed by using a synthetic dataset to assess: which method (RFR or SVR) has the best performance?; what the effect of using time, latitude and longitude as proxy variables is on  $\Delta pCO_2$ ?; and what is the impact of the sampling bias in the SOCAT v3 dataset is not yet fully representative of the Southern Ocean. The SVR was able to generalise better to the training dataset than the RFR with lower ratio between the out-of-sample and in-sample errors, but not enough to compensate for its poorer performance. The ensemble of the estimates show that interannual variability of the Southern Ocean  $CO_2$  sink is
- 20 dominated by the Polar Frontal Zone, while the Sub-Antarctic Zone is the dominant sink. on the estimates? We find that while RFR is indeed better than SVR, the ensemble of the two methods outperforms either one, due to complementary strengths and weaknesses of the methods. Results also show that for the RFR and SVR implementations, it is better two include coordinates as proxy variables as RMSE scores are lowered and the phasing of the seasonal cycle is more accurate. Lastly we show that there is only a weak bias due to undersampling. The synthetic data provides a useful framework to test methods in regions of
- 25 sparse data coverage and showing potential as a useful tool to evaluate methods in future studies.

**1 Introduction**

The global oceans have played an important role in mitigating the effects of climate change by taking up 25% of anthropogenic  $CO_2$  emissions annually (Khatiwala et al., 2013; Le Quéré et al., 2016). The Southern Ocean has played a disproportionate role in this uptake, accounting for 40% of the oceanic anthropogenic  $CO_2$  uptake (Khatiwala et al., 2013; Frölicher et al., 2015).

5 Yet, despite the region's importance, first order  $CO_2$  flux estimates are bound by large uncertainties due to sparse observations in the Southern Ocean (Lenton et al., 2006; Monteiro, 2010; Lenton et al., 2012; Takahashi et al., 2012; Bakker et al., 2016). These uncertainties limit our capacity to resolve variability and trends of  $CO_2$ .

Viable alternative methods to estimate net  $CO_2$  flux are atmospheric  $CO_2$  inversions, ocean biogeochemical process models and empirical models (Rödenbeck et al., 2015). As shown by Le Quéré et al. (2007), atmospheric  $CO_2$  inversions are useful

- 10 tools to estimate the net  $CO_2$  fluxes, but fail to offer further understanding with spatially integrated air-sea flux estimates (Fay and McKinley, 2014). Conversely, ocean biogeochemical process models are good tools for mechanistic understanding, but fail to represent seasonality of  $CO_2$  fluxes in the Southern Ocean (Lenton et al., 2013; Mongwe et al., 2016). Empirical modelling offers an opportunity to bridge the gap between sparse data in the Southern Ocean and correct parameterisation of future earth systems models.
- Empirical models maximise the utility of existing surface ocean  $CO_2$  observations ( $pCO_2$ ) by interpolating these with satellite proxy data. Access to in-situ  $pCO_2$  data, via platforms such as SOCAT (Surface Ocean  $CO_2$  Atlas), has been crucial to the success of empirical methods (Rödenbeck et al., 2015; Bakker et al., 2016). This, in conjunction with the increasing use of machine learning, has seen a proliferation in the number and diversity of methods in the literature. Rödenbeck et al. (2015) compared a suite of fourteen methods using a regional framework provided by Fay and McKinley (2014). The majority of these
- 20 methods are variants of multiple linear regression (MLR) or artificial neural networks (ANN), with regression being applied in regional windows or clusters based on climatologies of satellite measurable variables. The authors found that methods agreed in regions were where data coverage was adequate, but for data sparse regions, such as the Southern Ocean, interannual CO2 trends variability of various empirical methods were not coherent.

The primary reason for the varied results in Rödenbeck et al. (2015) is thought to be the way in which the algorithms deal

- 25 with sparse data in the Southern Ocean. These methods were typically variants of multiple linear regression (MLR) or artificial neural networks (ANN), with regression being applied in regional windows or clusters based on climatologies of satellite measurable variables. The SOM-FFN approach by Landschützer et al. (2014)exemplifies the combination of non-linear clustering coupled with regression. In a later work, Landschützer et al. (2015) used the SOM-FFN approach along with several other methods. Only two of the methods in Rödenbeck et al. (2015) were able to adequately represent interannual variability of
- 30  $\Delta pCO_2$ , namely: the SOM-FFN (self-organizing map feed forward neural network) from Landschützer et al. (2014), and the mixed layer scheme (MLS) from Rödenbeck et al. (2014). These two methods were used by Landschützer et al. (2015) to show that Southern Ocean CO2 uptake strengthened after 2000. However, these methods often showed large interannual differences in flux estimates despite agreeing on the overall decadal trend. This shows that there is lack of coherence even amongst the methods that perform well, meaning that different methods may lead to different interpretation of the lack of

measurements in the Southern Ocean meant that these methods could not be effectively tested with an independent dataset (Rödenbeck et al., 2015).

In the early 2000s, the North Atlantic experienced similar data paucity. Friedrich and Oschlies (2009) approached this problem by using process model output to evaluate the efficacy of an artificial neural network as well as finding the optimal

- 5 proxy variables for estimating *p*drivers of  $\Delta p$ CO2. This idealised environment was also used to estimate the effect of including/excluding certain proxy variables where it was found that filling remote sensing gaps in temperature and chlorophyll-a with climatology improved the estimates. In the intercomparison study by Rödenbeck et al. (2015) proxies typically include, but are not limited to : sea surface temperature (SST), chlorophyll-a (Chl-*a*), mixed layer depth (MLD) and sea surface salinity (SSS)The primary reason for the varied results is thought to be the way in which the algorithms deal with sparse data in the Southern Ocean
- 10 (Rödenbeck et al., 2015). This alludes to the importance of testing multiple approaches, as different methods may be able to better represent the CO2 estimates in the data sparse Southern Ocean.

In this study, we introduce and compare two empirical we introduce two methods new to this ocean  $CO_2$  application application, namely: Support Vector Regression (SVR) and Random Forest Regression (RFR). SVR is a method based on the theory of statistical learning, making the method robust to over-fitting by statistically determining the complexity of a problem rather than a

15 heuristic approach as required in setting up an ANNs hidden layer structure (Vapnik, 1999; ?). (Vapnik, 1999; Smola et al., 2004). In a review on the use of Support Vector Machines (the broad category for regression and classification variants) in remote sensing, (Mountrakis et al., 2011) found that the method had the "ability to generalize well even with limited training samples". This makes SVR an appealing consideration for the sparsely sampled Southern Ocean. RFR uses an ensemble of decision trees to create robust estimates, often without requiring data pre-processing making it an effective "off the shelf" method (Louppe, 2014).

As with SVM, Random Forests (both classification and regression variants) have also been used in remote sensing applications, though it does not seem to be as widely used in earth systems sciences despite proving to be a powerful, yet easy to implement, learning algorithm (Caruana and Niculescu-Mizil, 2006; Hastie et al., 2009). We use SVR and RFR to estimate  $CO_2$  fluxes in the Southern Ocean to try to better resolve the seasonal cycle from 1998 to 2014. These methods are trained with SOCAT

v3 data collocated with satellite proxies. We compare these results with those of Landschützer et al. (2014). In the next part we aim to better However, the lack of data in the Southern Ocean, particularly in winter, makes it difficult to understand the limitations of these methods within the framework of the SOCAT v3 data. context of SOCAT data.

To gain a better understanding of these methods' strengths and weaknesses we implement SVR and RFR are implemented in a simulated environmentwith a realistic sampling strategy to assess if there are biases to this sparse data . This approach allows

- 30 us to test the impact of including various in a synthetic data environment. A similar approach was taken by Friedrich and Oschlies (2009) in the North Atlantic, which experienced a similar data paucity to the Southern Ocean in the early 2000's. This idealised environment was also used to estimate the effect of including/excluding certain proxy variables as done by Friedrich and Oschlies (2009). Thereafter the methods are applied to observational data for actual estimates of p well as the optimal coverage of cruise tracks to constrain the North Atlantic  $\Delta p$ CO2 adequately. Similarly, we assess the efficacy of including coordinate variables as proxies
- 35 of  $\Delta p CO_2$  in the empirical methods. In the intercomparison study by Rödenbeck et al. (2015) proxies typically include, but are

**Table 1.** Information on data products used in this study. The temporal and spatial resolutions are for the raw data (before gridding). Dashes show that times are either not applicable or that the dataset is continually updated. Note that the start and end year show full years only. Links to download the data are given in the additional materials. The asterisk (\*) indicates that variables are the output of a data assimilative model.

| Group / Product | Variables               | Date Range |      | Resolution |                | Reference                                  |
|-----------------|-------------------------|------------|------|------------|----------------|--------------------------------------------|
| oroup / rroduce |                         | Start      | End  | Time       | Space          |                                            |
| SOCAT v3        | fCO2sea fCO2sea         | 1970       | 2014 | 1 mon      | 1°             | (Bakker et al., 2016)                      |
| CDIAC           | $\frac{xCO2atm}{xCO22}$ | 1970       | 2014 | -          | -              | ( ) ( Masarie et al., 2014 ) |
| Globcolour      | Chlorophyll             | 1998       | _    | 1 day      | $0.25^{\circ}$ | (Maritorena and Siegel, 2005)              |
| GHRSST          | Sea Surface Temperature | 1981       | _    | 1 day      | 0.25°          | (Reynolds et al., 2007)                    |
| ECCO2 (cube92)  | *Mixed Layer Depth      | 1992       | 2015 | 1 day      | 0.25°          | (Menemenlis et al., 2008)                  |
|                 | *Salinity               |            |      |            |                |                                            |

not limited to sea surface temperature (SST), chlorophyll-a (Chl-*a*), mixed layer depth (MLD) and sea surface salinity (SSS); however several methods in the study also include latitude and longitude. While coordinates do not mechanistically impact  $\Delta p CO_2$ , they do help to constrain estimates where the available remote sensing proxies cannot adequately do so. The synthetic data is also used to test the ability of the SVR and RFR to approximate  $\Delta p CO_2$  in the seasonally sparse Southern Ocean.

**5 2 Data and Methods**

This study is presented in two parts. The first applies SVR and RFR to the SOCAT v3 dataset and compares these outputs with those of the SOM-FFN by Landschützer et al. (2014). These estimates will be referred to as the observational estimates. Here the domain is limited to the three Southern Ocean (SO) domains of Fay and McKinley (2014) that are shown in Figure 1. These biomes are used to assess the performance of each of the methods, as done in Rödenbeck et al. (2015). Fay and McKinley

10 (2014) use a different nomenclature, which roughly corresponds to frontal zones. We rename the Sub-Tropical Seasonally Stratified biome (STSS) as the Sub-Antarctic Zone (SAZ); the Sub-Polar Seasonally Stratified biome (SPSS) becomes the Polar Frontal Zone (PFZ) and the ice biome (ICE) is the Antarctic Zone (AZ) (Mongwe et al., 2016).

The second part aims to better understand the limitations of these methods with the given dataset by implementing the methods to ocean biogeochemical model output. This will be referred to as the simulation experiment. Here the domain of the study

15 is south of 34°S – the biomes Fay and McKinley (2014). The domain of this synthetic data experiments is defined by the three southern biomes of Fay and McKinley (2014). These are defined by observed oceanographic and biological parametersand would thus be different in , but are used for the sake of consistency despite potential differences between observations and the model.

**Figure 1.** The three Southern Ocean biomes as defined by Fay and McKinley (2014). The common names for the biomes are shown in the key, with the abbreviations shown in the round brackets. The abbreviation in the square brackets show the abbreviations as given by Fay and McKinley (2014).

**2.1 Gridded Data**

The data sources are shown in Table 1. These gridded data refer primarily to remotely sensed data, with the exception of MLD and SSS. These The latter variables are output from  $ECCO_2$ , an assimilative modelspecific to the Southern Ocean. For the sake of brevity, these variables will be included under the description of "gridded observations". The temporal range of the

5 data (1998 through 2014) is limited by the availability of Globcolour (Chl-*a* starting in 1998) and SOCAT v3 (*f*CO2 ending in 2014).

All data are gridded to monthly x 1° using iris and xarray packages in Python (?Met Office)(Hoyer and Hamman, 2017; Met Office). Gridded  $pCO_2$  (SOCAT v3) is used to train the algorithms (Bakker et al., 2016). Surface station measurements (flask and tower) of atmospheric xCOxCO2 are interpolated to a regular grid using support vector regression (Masarie et al., 2014). Mean sea

10 level pressure (NCEP2) is used in the conversion from  $\frac{\text{xCO}_2 \text{CO}_2}{\text{xCO}_2}$  to  $p\text{CO}_2$  (Kanamitsu et al., 2002).

Cloud coverage and low light at high latitudes during winter result in missing Chl-*a* data. Cloud gaps are filled with the climatology of Chl-*a* (from 1998 to 2014) and missing low light data are filled with a value of  $0.1 \pm 0.03$  mg m-3 (uniformly distributed random noise).

**2.2 Model Data**

- 5 The prognostic coupled physics biogeochemical model used in this study is output from a regional NEMO-PISCES configuration , (BIOPERIANT05-GAA95b. This model) is used as the synthetic dataset. The configuration is an updated version of PERIANT05 used by Dufour et al. (2012), where BIOPERIANT05-GAA95b includes biogeochemistry with PISCES-v2. The model has a peri-Antarctic domain with an open northern boundary at 30°S. The horizontal resolution of the configuration is 0.5° cos(latitude) with 46 vertical levels. The northern boundary is forced by a global 0.5° model, ORCA05 as presented
- 10 in Biastoch et al. (2008). Output was is saved as five-day averages. The simulation was run from 1992-1998 to 2009. The synthetic observations are sampled at the model resolution (5-day  $\times 0.5^{\circ}$ ) to resemble the SOCAT dataset. Hereafter all data is resampled to 1.0° spatial resolution and monthly temporal resolution data to match observations. Finally, for the simulation experiment we define the Southern Ocean using the three southernmost biomes defined in Fay and McKinley (2014) as done for the observational estimates.

**15 2.3 Data transformation and derived variables**

There are several transformations that are applied to data for both model output and gridded observations. Both gridded data and synthetic input data are transformed in preparation for the empirical algorithms. The  $log_{10}$  transformations of MLD and filled chlorophyll (Chl- $a_{clim}$ ) are taken to return a distribution that closer represents a normal distribution.

- Several of the studies in Rödenbeck et al. (2015) included latitude, longitude and/or time as proxies of Δ*p*CO2. However,
  many of the methods that are regional or cluster the data before regression. It is important to note that coordinates do not drive mechanistic changes in ΔpCO2. Rather, the inclusion of coordinates in the empirical methods account for unknown or regionally varying proxies that cannot be measured remotely. Many methods in the intercomparison by Rödenbeck et al. (2015) did not include coordinates, but account for unaccountable spatial variability by clustering or subsetting data regionally. In this study, we use a single large domain with no clustering or regional subsets. This then raises the question of whether including
- 25 coordinates would improve estimates or not. Including the coordinates may create a model where the training location is too narrow. Two scenarios for each method in the simulation experiment are run: no coordinate variables, and including coordinate variables (time, latitude and longitude).

The coordinates are transformed to preserve the continuity of the data as is shown below. Seasonality of the data is preserved by transforming the day of the year (j) and is included in both SVR and RFR analyses:

$$\quad t = \begin{pmatrix} \cos\left(j \cdot \frac{2\pi}{365}\right) \\ \sin\left(j \cdot \frac{2\pi}{365}\right) \end{pmatrix} \tag{1}$$

Transformed coordinate vectors were passed to only SVR both SVR and RFR using n-vector transformations of latitude ( $\lambda$ ) and longitude ( $\mu$ ) (Gade, 2010; Sasse et al., 2013), with n containing:

$$A, B, C = \begin{pmatrix} \sin(\lambda) \\ \sin(\mu) \cdot \cos(\lambda) \\ -\cos(\mu) \cdot \cos(\lambda) \end{pmatrix}$$
(2)

Co-located fCO2 (y) and proxy data (X) were are used to create training arrays (x). The final input for SVR were the **following proxies and RFR are** (with 12 columns):  $\log_1 0$ (Chl- $a_{clim}$ ), SST,  $fCO_{2(atm)}$ , ADT,  $\log_{10}$ (MLD), ICE, SSS,  $\cos(j)$ ,  $\sin(j)$  and n-vectors [A, B, C]. SVR requires each column of the proxies to be z-scored; i.e. normalized to the mean ( $\mu$ ) and standard deviation ( $\sigma$ ) of each column ( $\frac{x-\mu}{\sigma}$ ).

**2.4 Empirical methods and implementation**

Data is- Data are split randomly into a training and independent test dataset with a ratio 0.7:0.3. The independent dataset
is used to give a test error of the trained algorithm. The statistical learning package, Scikit-Learn, in Python is used for all regression and cross-validation methods (Pedregosa et al., 2011). The details on each cross-validation method are outlined in the subsections below.

**2.4.1 Support vector regression**

The basic formulation of SVR is such that similar to that of linear regression as described by Smola et al. (2004):

15
$$f(x) = \langle w, \mathbf{x} \rangle + b$$
 with  $b \in \mathbb{R}$  (3)

where b is an intercept,  $\langle \cdot, \cdot \rangle$  denotes the dot product of the cost function minimizes the number of points on or outside the allowable error margins (c) as shown in 2a. A few slack variables ( $\xi$ ) are allowed, within the limits of a slack parameter (C), which is set by the user. The weights (w) and x, the training data. The weights and intercept are found by solving the cost function:

20
$$\underbrace{\min initial constraints}_{2} \frac{1}{2} ||w||^2 \text{ subject to} \begin{cases} y_i - \langle w, x_i \rangle - b & \leq \epsilon \\ \langle w, x_i \rangle + b - y_i & \leq \epsilon \end{cases}$$
(4)

In this form, w is minimised according to the target values  $(y_i)$  to a precision of  $\epsilon$  – i.e. there is no room for error greater than  $\epsilon$ . However, with the majority of problems, meeting these constraints is not possible if data are noisy or  $\epsilon$  is set small. The inclusion of slack variables  $(\xi_i, \xi_i^*)$  relaxes the constraints and the problem is now formulated as:

$$\underbrace{\min_{i=1}}_{\substack{i=1\\ \dots \dots \dots \dots}} \underbrace{\frac{1}{2} ||w||^2 + C \sum_{i=1}^n (\xi_i + \xi_i^*)}_{\underset{i=1}{\dots \dots}} \qquad \underbrace{\operatorname{subject to}}_{\underset{i=1}{\dots}} \begin{cases} y_i - \langle w, x_i \rangle - b & \leq \epsilon + \xi_i \\ \langle w, x_i \rangle + b - y_i & \leq \epsilon + \xi_i^* \\ \xi_i, \xi_i^* & \geq 0 \\ \underset{\underset{i=1}{\dots \dots}}{\dots \dots} \end{cases}$$
(5)

Here C is a parameter that adjusts for the amount of error that the minimisation allows. The slack variable  $|\xi|$  is only counted towards the cost if the point lies outside the margin ( $|\xi| \ge \epsilon$ ). The points on or outside these margins are the the margins are called support vectors and are used to construct the hypothesis function, h(x). This elegant approach is made versatile by mapping X is shown in Figure 2a where a linear SVR is fitted to noisy data produced from a cubic spline. The optimisation

5 problem shown in Eq. 5 is solved in its dual formulation (see Hastie et al. 2009 for the full description). Importantly, solving the dual formulation allows for efficient kernelisation of SVR.

Kernelisation describes the process that maps the proxy variables (x) onto a higher dimensional feature spaceusing an interchangeable kernel. In this study we used a Gaussian kernel (or radial basis function – RBF), which allows for potentially infinite complexity - determined by the number of support vectors (Vapnik, 1999). The assignment of the number of support

10 vectors is analogous to defining the architecture of an ANN. The RBF kernel introduces an additional hyper-parameter ( $\gamma$ ) that defines the width of the Gaussian. Selection of the SVR hyper-parameters ( $\epsilon$ , C,  $\gamma$ ) is done using a two-stage coarse-fine exhaustive grid search approach using with cross validation. We use K-fold cross validationwith, where the data is divided into eight equal "folds" (k = 8-). Seven of the folds are used to train the model, while the remaining fold is used for validation. This is done iteratively until each excluded fold has been used to test the results.

**15 2.4.2 Random Forest Regression**

A random forest-Decision trees form the basic building block of a Random Forest (RF)is an ensemble of decision trees, which means that the average estimate of n trees is taken, with the average of n decision trees is taken as the ensemble estimate (Breiman, 2001) (Figure 2b). The basic idea of a decision tree is to iteratively partition data into boxes using simple rules that minimize the error at each split (referred to as a node) – these boxes would become hypercubes in higher dimensional

- 20 problems. This is described by the basic formulation as described in Loh (2011):
  - 1. Start at the root node
  - 2. For each X, find the set S that minimizes the sum of the node impurities in the two child nodes and choose the split  $X \in S$  that gives the minimum overall X and S.
  - 3. If a stopping criterion is reached, exit. Otherwise, apply step 2 to each child node in turn.
- 25 Decision trees have high variance due to their discrete nature. Random forests reduce the high variance of decision trees by bagging (bootstrap aggregating)in which the this high variance by bootstrapping with aggregation (called bagging): a subset of the available training dataset is sampled with replacement resulting in for each decision tree in the RF. The sampling with replacement means that each training observation has a  $\sim 63\%$  chance of being chosen at least once for a particular tree (Louppe, 2014). A This subsampling provides estimates that are robust to outliers as these have a chance of being omitted in
- 30 training. This means that a random forest typically performs better when number of decision trees (t) is large, but increasing the number of trees has diminishing returns in terms of performance vs. computation. Additional robustness is given to RFs by randomizing and/or limiting the number of proxy variables (m) given to the nodes in each tree when splitting the data

Figure 2. A simple example demonstrating the principle of (a) support vector regression and (b) random forest regression. The dashed grey line is the true function  $f(x) = 0.4x^3$  with the blue dots representing a random sample taken from this function  $f(x)+\sigma$ , where  $\sigma$  is normally distributed noise. The black line in each figure, h(x), show the estimate of the true function. The orange dots in (a) show the samples from the random subset chosen as support vectors from which h(x) is estimated. The orange lines in (b) show 200 decision tree estimates,  $g_i(x)$ , which are averaged to create the ensemble, h(x).

(hence random) (Louppe, 2014). In this study, the maximum number of proxy variables (m = 11) was given to the RFR. The complexity of a RF can be adjusted by limiting the minimum number of leaves at a terminal branch (l), where a fully-grown tree would allow l to be one; tree depth can also be limited to reduce the complexity and has a similar effect to limiting l.

5

A useful feature of bagging is that it intrinsically provides a cross-validation dataset (a.k.a. out-of-bag samples) that is not part of the training procedure (for a specific set of trees). The out-of-bag samples are those that are not selected during bagging. The advantage of this approach over K-fold cross-validation is that the full dataset can be used in the training procedure, as opposed to splitting the dataset for cross-validation. The out-of-bag error is used to cross-validate the model and select the hyper-parameters (t, m, l) for the RF.

---

## Referee Report (RR1)

**2nd review of Gregor et al: Empirical methods for the estimation of Southern Ocean CO2: Support Vector and Random Forest Regression. Submitted to Biogeosciences**

**Response to previous concerns:**

In my previous assessment, I have raised 3 main concerns. The first related to the many new terms that are not explained in the methods section, the second related to the lack of validation and assessment of the results and the third related to the unknown consequences of adding geographical proxy data.

The authors have substantially revised their methods section. It is now much easier to understand how random forest and support vector regression work. The authors did an excellent job explaining all previously undefined wording. As previously mentioned: Understanding a method means trusting a method. I have now much more trust in both methods and the results of the manuscript. Hence, I have no further reservations regarding the methods section.

The authors have further added additional validation of the methods in the revised manuscript. The authors added both the temporal and spatial error in Figure 5 as well as a sectoral error analysis in table 2. This adds additional confidence. Therefore, I have no more reservations regarding the validation of the results.

Lastly, the author also addressed my last point of concern. Namely, the discussion regarding the inclusion of coordinates as proxy variables. Despite the authors describing the changes made in their response to reviewer letter as "addressed a little better", I would even go a step further and say this is "addressed much better". I however have found the paragraph in the introduction – not section 2.3 - but that is of no concern (in fact I think the paragraph should stay where it is)

In summary, I am happy to say that the authors have adequately addressed all my previously raised concerns and substantially improved their manuscript.

**Recommendation:**

Based on the revision, I am happy to **recommend the manuscript for publication in Biogeosciences**. I have gathered a few technical points below that I would like the authors to consider before publication. Some of them editorial, i.e. concerning spelling etc. and some concerning the wording used.

**Specific comments:**

Page 1 Line 5, page 17 line 9 and page 21 line 10: I believe the authors are too negative here. Based on the evidence presented, I would not say the SOM-FFN "outperforms" the other methods. This is also along the lines of a comment made by reviewer 2 in the first round of reviews. The RMSE values are so similar and close that I would not say one method is better than the other, but rather say they "depict errors of similar magnitude".

Page 1 line 17: "Lastly, …"

Page 2 line 32: "… sensing, Mountrakis et al., 2011 found …"

Page 12 line 5: "Importantly, …"

Page 18 line 25: "(Figure 6 b,c)"

Page 21 line 4: I would certainly say they are complementary.

In general: Map plots seem to miss the last (or first) longitude entry (hence the white stripe). It often helps to plot the array 2x in longitude, i.e. from 0-720 degrees rather than 0-360 degrees. Then the stripe does not appear.

---

## Author Response (AR2)

We would like to thank reviewer R1 for reassessing the manuscript. I have added the responses below each of the questions.
I have also attached the track changes below these responses.

**2nd review of Gregor et al: Empirical methods for the estimation of Southern Ocean CO2: Support Vector and Random Forest Regression. Submitted to Biogeosciences**

**Response to previous concerns:**

In my previous assessment, I have raised 3 main concerns. The first related to the many new terms that are not explained in the methods section, the second related to the lack of validation and assessment of the results and the third related to the unknown consequences of adding geographical proxy data.

The authors have substantially revised their methods section. It is now much easier to understand how random forest and support vector regression work. The authors did an excellent job explaining all previously undefined wording. As previously mentioned: Understanding a method means trusting a method. I have now much more trust in both methods and the results of the manuscript. Hence, I have no further reservations regarding the methods section.

The authors have further added additional validation of the methods in the revised manuscript. The authors added both the temporal and spatial error in Figure 5 as well as a sectoral error analysis in table 2. This adds additional confidence. Therefore, I have no more reservations regarding the validation of the results.

Lastly, the author also addressed my last point of concern. Namely, the discussion regarding the inclusion of coordinates as proxy variables. Despite the authors describing the changes made in their response to reviewer letter as "addressed a little better", I would even go a step further and say this is "addressed much better". I however have found the paragraph in the introduction – not section 2.3 - but that is of no concern (in fact I think the paragraph should stay where it is)

In summary, I am happy to say that the authors have adequately addressed all my previously raised concerns and substantially improved their manuscript.

**Recommendation:**

Based on the revision, I am happy to **recommend the manuscript for publication in Biogeosciences**. I have gathered a few technical points below that I would like the authors to consider before publication. Some of them editorial, i.e. concerning spelling etc. and some concerning the wording used.

**Specific comments:**

Page 1 Line 5, page 17 line 9 and page 21 line 10: I believe the authors are too negative here. Based on the evidence presented, I would not say the SOM-FFN "outperforms" the other methods. This is also along the lines of a comment made by reviewer 2 in the first round of reviews. The RMSE values are so similar and close that I would not say one method is better than the other, but rather say they "depict errors of similar magnitude".
 I have addressed this issue by rewording as marked in the text

Page 1 line 17: "Lastly, ..."
 Comma inserted

Page 2 line 32: "… sensing, Mountrakis et al., 2011 found …"
Changed the brackets - now reads Mountrakis et al., (2011)

Page 12 line 5: "Importantly, …"
Comma inserted

Page 18 line 25: "(Figure 6 b,c)"
Changed the bracket

Page 21 line 4: I would certainly say they are complementary.
The statement is now more assertive

In general: Map plots seem to miss the last (or first) longitude entry (hence the white stripe). It often helps to plot the array 2x in longitude, i.e. from 0-720 degrees rather than 0-360 degrees. Then the stripe does not appear.
This has been fixed! Solved the issue by changing the longitude to extent from -180° to 181°.
Thanks for the tip

**Empirical methods for the estimation of Southern Ocean $CO_2$: Support Vector and Random Forest Regression**

Luke Gregor[1,2], Schalk Kok[3], and Pedro M. S. Monteiro[1]

[1]Southern Ocean Carbon-Climate Observatory (SOCCO), CSIR, Cape Town, South Africa
[2]University of Cape Town, Department of Oceanography, Cape Town, South Africa
[3]University of Pretoria, Department of Mechanical and Aeronautical Engineering, Pretoria, South Africa

*Correspondence to:* Luke Gregor (luke.gregor@uct.ac.za)

**Abstract.** The Southern Ocean accounts for 40% of oceanic $CO_2$ uptake, but the estimates are bound by large uncertainties due to a paucity in observations. Gap filling empirical methods have been used to good effect to approximate $pCO_2$ from satellite observable variables in other parts of the ocean, but many of these methods are not in agreement in the Southern Ocean. In this study we propose two additional methods that perform well in the Southern Ocean: Support Vector Regression (SVR) and Random Forest Regression (RFR). The methods are used to estimate $\Delta pCO_2$ in the Southern Ocean based on SOCAT v3, achieving similar trends to the SOM-FFN method by Landschützer et al. (2014). Results show that the SOM-FFN  and RFR approaches have RMSEs of similar magnitude (14.84  and 16.45 µatm) where the SVR method has a larger RMSE (24.40 µatm). However, the larger errors for SVR and RFR are, 
[revised manuscript text omitted]